# Minimally invasive steam-assisted drug delivery with ICG fluorescence guidance for primary malignant bone tumors and evaluation of clinical applicability

Seon Min Lee[1,2‡], Kicheol Yoon[3‡], Sangyun Lee[4‡], Hyun Guy Kang[5*], Kwang Gi Kim[1,2,3,6,7*]

**1** Department of Biomedical & Bio-Health Medical Engineering, Gachon Advanced Institute for Health Sciences and Technology (GAIHST), Gachon University, Republic of Korea, **2** Medical Devices R&D Center, Gachon University Gil Medical Center, Incheon, Republic of Korea, **3** Gachon Biomedical Convergence Institute, Gachon University Gil Medical Center, Incheon, Republic of Korea, **4** Department of Radiological Science, Dongnam Health University, Suwon, Republic of Korea, **5** Orthopaedic Oncology Clinic, Center for Rare Cancer, National Cancer Center, Goyang, Republic of Korea, **6** Department of Biomedical Engineering, Gachon University, Seongnam, Gyeonggi-do, Republic of Korea, **7** KMAIN Corp., 621, 622, Business Growth Center, Seongnam, Gyeonggi-do, Republic of Korea

‡ These authors are co-first authors on this work.
* ostumor@ncc.re.kr (HGK); kimkg@gachon.ac.kr (KGK)

## Abstract

### Background

Malignant bone tumors are rare cancers with a poor prognosis, often causing severe pain and pathological fractures that substantially reduce patients' quality of life. Conventional cemento-plasty can inhibit tumor cell proliferation but is limited by insufficient drug diffusion. To address this, we developed a minimally invasive drug delivery system utilizing high-temperature, high-pressure steam with integrated sensing and real-time temperature monitoring.

### Methods

A drug delivery system consisting of a water tank, pump, and steam generator was designed and fabricated. The system was equipped with a monitoring unit capable of real-time temperature measurement and control.

### Results

Large-animal experiments were conducted to evaluate the feasibility and distribution of steam injection in the femur. The procedure increased bone site temperatures to 48.8°C, as confirmed by fluorescence imaging.

### Discussion

High-temperature steam successfully reached the target tissue; however, additional research is required to minimize collateral damage to normal bone. Future studies

**Data availability statement:** All relevant data are within the paper and its Supporting Information files.

**Funding:** This work was supported by research funding from the Korea Evaluation Institute of Industrial Technology (KEIT) in the form of a grant (RS-2025-02305698) received by KGK. No additional external funding was received for this study.

**Competing interests:** The authors have declared that no competing interests exist.

should focus on enhancing practicality by refining steam pressure and temperature control, optimizing nozzle design, and miniaturizing the device.

## Conclusion

The proposed steam-based drug delivery system achieved broader tissue distribution than conventional methods, demonstrating potential as a novel treatment strategy for metastatic bone tumors. Further preclinical studies are warranted to support its clinical translation.

## Introduction

Malignant bone tumors are rare cancers with a poor prognosis, with reported 5-year and 10-year survival rates of 65.3% and 58.0%, respectively, in the United States in 2024 [1]. Since the present study was conducted in Korea, U.S. survival rate statistics were referenced to enable international comparison. Metastatic bone tumors the most common type of malignant bone tumor cause severe pain, gait disturbance, and pathological fractures [2,3]. In contrast, this study focuses primarily on primary malignant bone tumors, which must be clearly distinguished from metastatic tumors. Although metastatic bone tumors are the most common overall, primary bone tumors (e.g., osteosarcoma) are much rarer and exhibit distinct clinical characteristics and treatment approaches.

Pain associated with malignant bone tumors can be alleviated through cementoplasty and thermal ablation [3–5]. Cementoplasty is primarily used for pain relief and bone stabilization and has been shown to prevent bone hardening and fractures. However, this study seeks to demonstrate the necessity and rationale for developing more effective alternatives to cementoplasty. Advances in surgical techniques now allow more than 90% of extremity sarcomas to be treated without amputation, enabling tumor removal without injury to major nerves and blood vessels. As long as sufficient soft tissue, including skin, remains, virtually all sarcoma cases are eligible for limb-sparing surgery [6].

In this surgical approach, the tumor-affected bone is extensively excised or destroyed, and the resulting defect is reconstructed. Reconstruction techniques often utilize artificial substitutes such as metallic implants [7,8]. The survival rates of metallic implants average 82% at 5 years and 75% at 10 years [9]. However, implants may damage surrounding tissues and are associated with complications such as infection, aseptic loosening, and fracture [5,7,10,11]. In pediatric patients, implants must be customized to account for growth plates, which further increases treatment costs.

Cementoplasty offers another option for defect reconstruction, in which a hole is drilled directly into the bone for injection of drugs or bone cement [12,13]. While cementoplasty can suppress tumor cell proliferation, its therapeutic coverage is limited to localized areas [6]. To overcome this limitation, drug particle diffusion can be enhanced by applying pressure. Delivering the drug in the form of steam provides the additional advantage of rapid diffusion, owing to the high absorption rate of

porous tissue and the small particle size. This approach results in superior therapeutic performance compared with liquid formulations [14]. Recent advances in cancer research have highlighted the critical role of immune evasion mechanisms in tumor progression [15]. Specifically, when PD-L1 expressed on the surface of tumor cells binds to the PD-1 receptor on T cells, the cytotoxic activity of T cells is suppressed, allowing tumor cells to evade immune surveillance and survive. This pathway plays a pivotal role in sustaining tumor growth, and immune checkpoint inhibitors targeting this mechanism have demonstrated significant therapeutic efficacy across multiple cancer types [15].

Recently, high-temperature, high-pressure steam-based treatment devices have been increasingly applied to cancer tissue ablation, anticancer drug delivery, and infection control. However, conventional steam generators often exhibit fluctuations in steam temperature during treatment, which can compromise both therapeutic efficacy and safety [16]. Specifically, overheating increases the risk of damage to normal tissue, whereas insufficient heating reduces therapeutic effectiveness [17]. Maintaining a stable target temperature is therefore a critical requirement for steam-assisted drug delivery systems in cancer therapy [18]. Although rapid heating mechanisms have been investigated, temperature fluctuations during treatment remain a major challenge, potentially undermining both efficacy and safety [18]. To address this limitation, a control circuit with real-time temperature monitoring and heating regulation has been developed, providing visual warnings for overheating, insufficient heating, and optimal treatment conditions [19].

In this study, steam generation was integrated with a control circuit that enables continuous temperature monitoring, stepwise warning signals, and precise regulation. The module adjusts the heater output based on real-time sensor data and provides intuitive status indications through color-coded LEDs (red, yellow, and green), thereby ensuring stable steam conditions throughout the procedure. Furthermore, the proposed high-temperature, high-pressure steam-based anticancer drug delivery system is expected not only to enhance drug diffusion but also to induce tumor cell necrosis via hyperthermia and to modulate the immunosuppressive tumor microenvironment (TME). These effects may provide secondary benefits, including restoration of T cell activity and increased tumor antigen exposure, thereby promoting an antitumor immune response.

In addition, applying the principle of thermal ablation to high-temperature drug injection can achieve more effective inhibition and destruction of tumor cells [3,14]. This paper proposes the design of a steam-based system capable of minimally invasive anticancer drug delivery using high temperature and high pressure. The primary objective of this study was to develop a novel anticancer drug injection device capable of delivering high-temperature, high-pressure steam. The device's injection performance and safety were evaluated through preclinical animal testing using healthy bone tissue. It is important to note that these experiments did not target cancerous tissue directly; rather, they assessed the system's performance by analyzing the diffusion and injection effects of steam within healthy bone.

Therefore, this study focuses on elucidating the potential and physical characteristics of high-temperature, high-pressure steam injection technology rather than directly evaluating clinical therapeutic efficacy. Accordingly, rather than comparing this technology directly with existing clinical treatments, the goal was to develop an innovative approach to overcome the diffusion limitations inherent in procedures such as cementoplasty.

## System manufacturing and drug delivery methods

The conventional treatment for bone tumors involves drilling a hole into the femur (diameter $D_2 = 6.0\,mm$, length $L_2 = 16\,mm$), followed by the insertion of a screw (diameter $D_1 = 3.0\,mm$, length $L_1 = 13\,mm$), as illustrated in Fig 1 A. Subsequently, an 11-gauge (11G) needle, matched to the drilled hole diameter, is used to inject the drug in liquid form [13].

The injected drug binds to tumor cells and induces cell death, as illustrated in Fig 1B [20]. Cancer cells express an immune-evasive protein, Programmed Death-Ligand 1 (PD-L1), which suppresses immune cell function. When PD-L1 on cancer cells binds to PD-1 receptors on T cells, the immune response is suppressed, enabling cancer cells to evade immune surveillance and avoid destruction, as illustrated in Fig 1B [13].

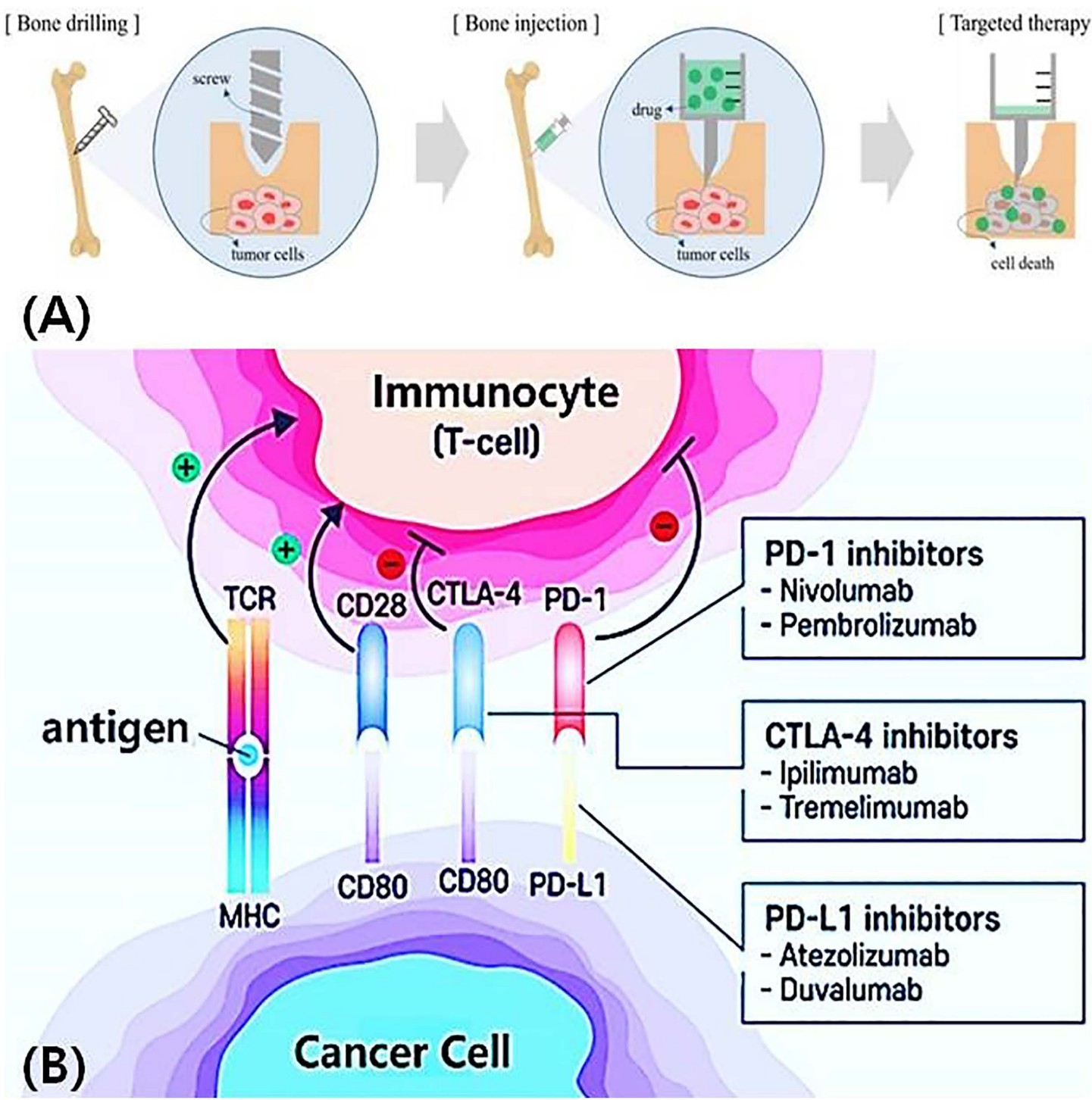

**Fig 1. Conventional Bone bone tumor treatment and immune evasion mechanism via pd-1/pd-l1 pathway.** (A) conventional treatment of bone tumors using drilling and injecting anticancer drugs (B) mechanism of immune evasion via pd-1/pd-l$_i$ interaction.

If administered PD-1 antibodies bind to the interaction site between PD-L$_1$ and PD-1, the immune evasion signal is blocked, enabling T cells no longer suppressed to induce necrosis in cancer cells [20,21]. However, because immune checkpoint inhibitors can cause autoimmune side effects affecting multiple organs, including the digestive, respiratory, circulatory, endocrine, and nervous systems, continuous monitoring throughout the treatment period is essential. To address these challenges, the proposed system is designed to deliver targeted therapy by heating the drug to generate steam, as illustrated in Fig 2, and directly injecting the steam into the local bone area corresponding to the malignant tumor.

The system operates on a 220 V power supply and incorporates an internal relay switch, which controls the current flow from the power supply to both the water tank and the steam generator. The water tank is filled with 100 mL of 0.9% normal saline (NS), which is delivered to the water pump. The steam generator heats the NS to a high temperature, and the heated saline is then sprayed through the steam valve and nozzle, propelled by a pressure motor, toward the lesion site (simulated). Accordingly, the drug is sprayed over a bone area of approximately 26.6 cm², following the principle of tumor ablation by heat. The saline in the water tank is heated to 120°C by the steam generator and injected through the needle at a flow rate (Fs) of 50 mL/min and a pressure of 1.97 atm, as illustrated in Fig 3A and described by Eq 1 [22].

$$T_x = T_n + (T_{int} - T_n)e^{-\frac{hz}{\rho c \rho v}}$$

(1)

The drug (normal saline, NS), heated by the steam generator and stored in the water tank, travels through the nozzle to the needle tip, as illustrated in Fig 3B and Fig. 3C. The nozzle has a length of 14.6 cm, and the drug travel time is approximately 6 s. The nozzle conveys the heated drug from the steam generator to the lesion site, which is located 14.6 cm away, within about 6 s. The high-temperature drug is then delivered to the lesion through the needle for 7 min, allowing the

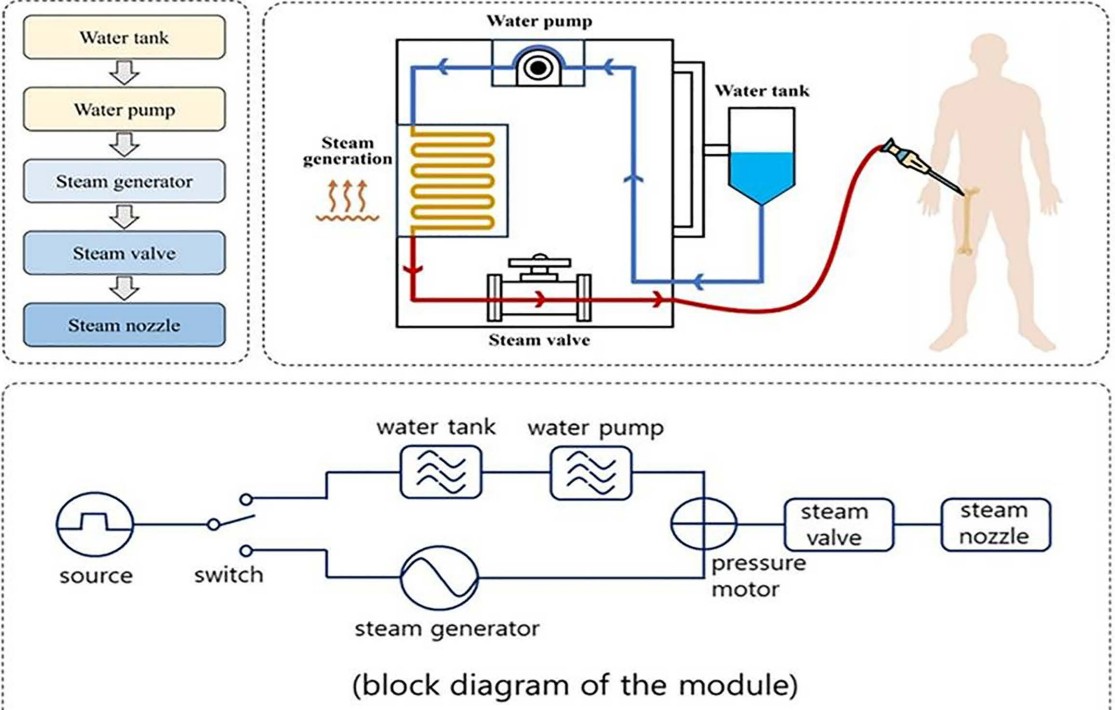

**Fig 2. Flowchart, structure, and block diagram of the behavior of a system design.**

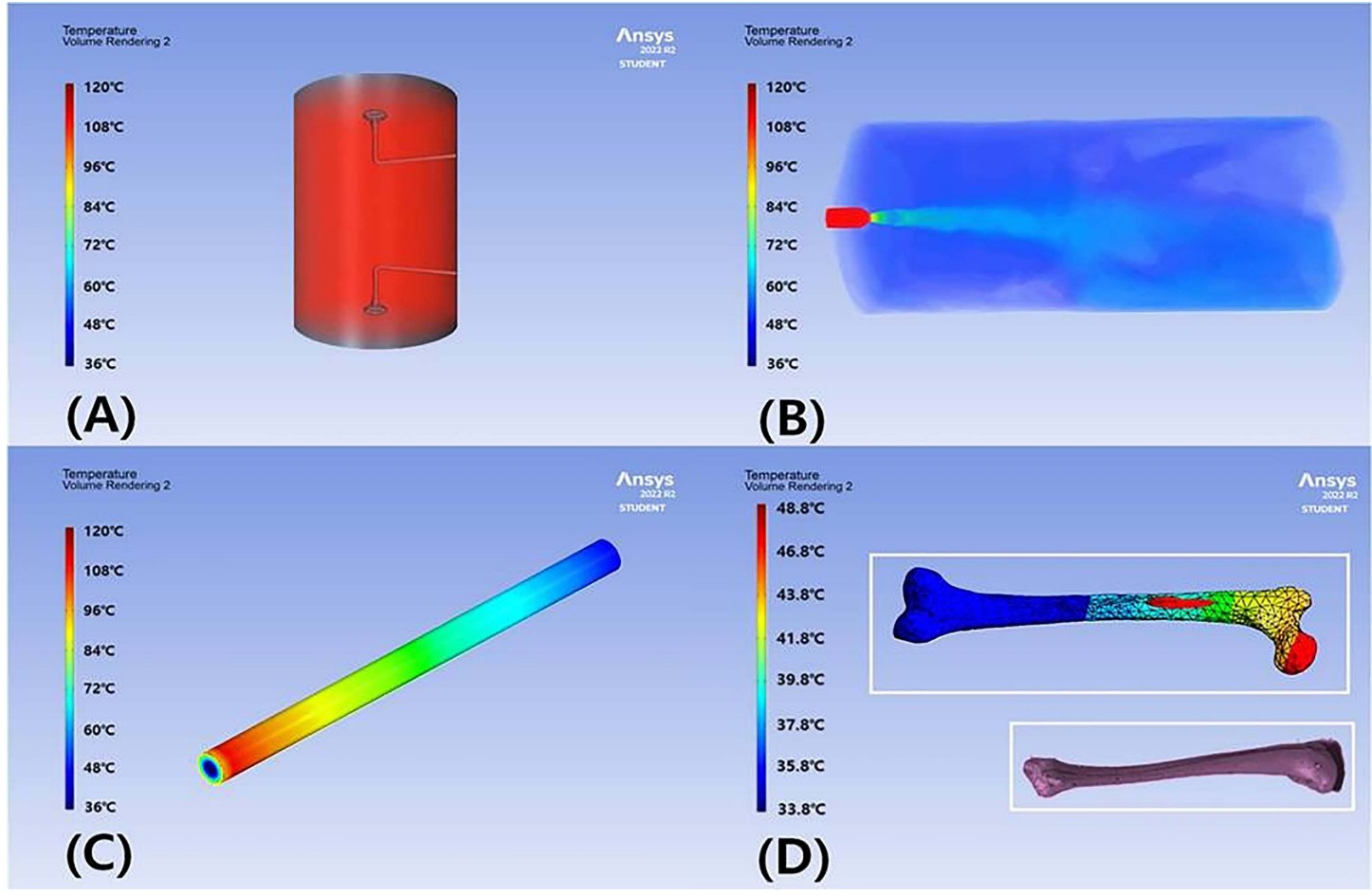

**Fig 3. Simulation results results of temperature change for high-temperature ns 0.9%.** (A) schematic of the steam-heated saline delivery system (water tank, steam generator, nozzle, and needle) (B) Injection injection of heated normal saline (ns) through the needle at a flow rate ($f_s$) of 50 ml/min under 1.97 atm (C) transit of the heated saline from the steam generator to the needle tip through a 14.6 cm nozzle (≈6 s) (D) thermal delivery to the bone for 7 min via an 11-gauge (11g) needle.

lesion site (x) to be thermally treated, as shown in Fig 3D. During this process, steam is injected via an 11G needle, which is inserted into the lesion (bone) according to the established procedure.

The flow rate (Q) of the drug through the nozzle is 50 mL/min, corresponding to a flow velocity (v) of $8.33 \times 10^{-7}$ m³/s, as shown in Eq 2. The nozzle diameter (n) is 3 mm, resulting in a cross-sectional area (A) of $7.07 \times 10^{-6}$ m², as shown in Eq 3 [22].

$$v = \frac{Q}{A} \tag{2}$$

$$A = \pi \left(\frac{n}{2}\right)^2 \tag{3}$$

Assuming NS, the drug heated to 120°C in the water tank by the steam generator experiences a temperature decrease according to Newton's law of cooling, with a factor of 0.407, yielding an average temperature of 15.1°C during travel to the needle tip (x), as shown in Eq. 4 and Fig 3C [22].

$$T_n \cong \frac{T_f - T_i e^{-kt}}{1 - e^{-kt}} \tag{4}$$

The constant k in Eq 4 ranges from 0.15 to 0.3 s$^{-1}$. Since k depends on the nozzle and its heat transfer characteristics, it was determined to be 0.22 s$^{-1}$. For the analysis of k and flow velocity v, the heat transfer coefficient (h), density ($\rho$), and specific heat capacity at constant pressure (Cp) were set to 50 W/m²·K, 1000 kg/m³, and 4180 J/kg·K, respectively.

Based on this analysis, the temperature $T_x$ of the drug traveling from the water tank to the needle through the nozzle, as well as the temperature of the drug sprayed from the needle, was calculated to be 48.8°C, as shown in Eq 5. At this temperature, the high-temperature drug (assuming NS) reaches the lesion site. The results of these analyses are summarized in Table 1 [22].

$$T_x = T_n + (T_{int} - T_n)e^{-kt} \tag{5}$$

The steam generator is connected to the nozzle via an insulated pipe, and a control device based on an MCU board is configured to maintain the target temperature using hysteresis control, as described by Eq 6.

$$P(t) = \begin{cases} 0 & T(t) > T_{target} + \Delta T \\ 1 & T(t) < T_{target} - \Delta T \\ P(t-1) & others \end{cases} \tag{6}$$

Here, $P_{(t)}$ represents the heater drive signal (1: ON at 120°C, 0: OFF), $T_{(t)}$ is the real-time measured temperature, and $\Delta T$ denotes the allowable temperature deviation. According to the equation, if the temperature exceeds the upper limit (>121°C), the heater is turned off. Conversely, if the temperature falls below the lower limit (<119°C), the heater is turned on. Within the intermediate range, the previous state is maintained at 120°C to prevent unnecessary switching. While automatic ON/OFF control is implemented, manual operation is also feasible for clinical applications; however, continuous monitoring remains essential.

The temperature sensor, shown in Fig 4, is a Type-K thermocouple with an accuracy of ±0.1°C, positioned 2 cm upstream of the nozzle. Visual indicators include a green LED for the optimal temperature range (~120°C), a yellow LED for the low temperature range (<119°C), and a red LED for the overheated range (>121°C). Temperature control is relay-based, with the heater automatically shutting off when the upper temperature limit is exceeded and restarting when the temperature falls below the lower limit.

**Table 1. Thermal and fluid characterization of drug delivery through nozzles.**

| performance | parameter | performance | parameter |
|---|---|---|---|
| water tank | 100 mL | P | 1000 kg/m³ |
| $F_s$ | 50 mL/min | $C_p$ | 4180 J/kg·K |
| $T_{int}$ | 120°C | v | $8.33 \times 10^{-7}$ m³ |
| P | 1.97 atm | n | 3 mm (= 0.003 m) |
| L | 14.6 cm | A | $7.07 \times 10^{-6}$ m² |
| injection time | 6 sec | k | 0.15 — 0.3 s$^{-1}$ |
| treatment duration | 7 minutes | selected k value | 0.22 s$^{-1}$ |
| needle gauge | 11G | $T_x$ | 48.8°C |
| $T_n$ | 15.1°C | temperature reduction ratio | Decreased by 0.407 times |
| h | 50 W/m²·K | | |

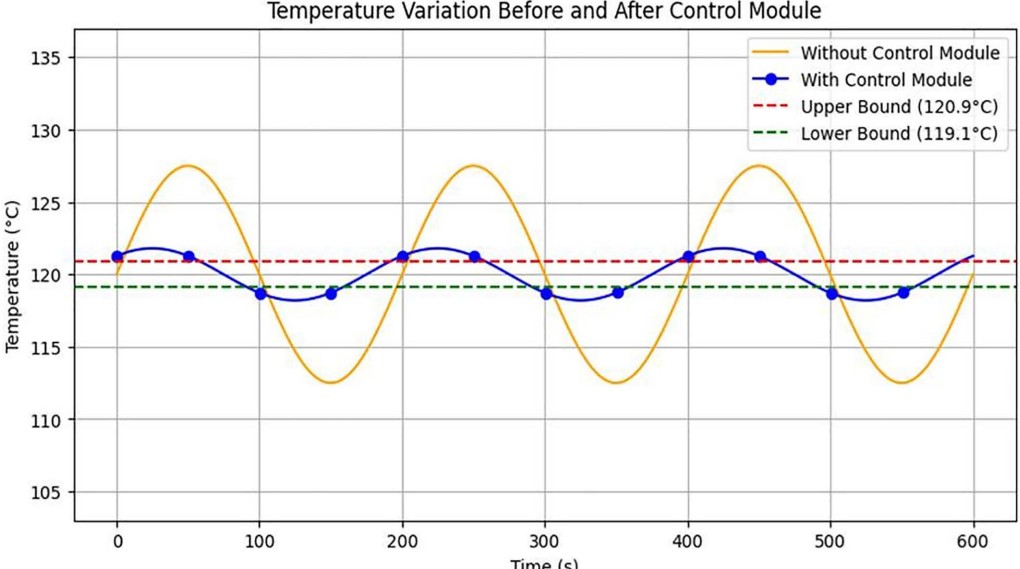

**Fig 4. Temperature change change simulation results for heater heat generation.**

Temperature measurements were performed at a frequency of 1 Hz for 10 min under two conditions: with the control module OFF and with the control module ON. Temperature stability was evaluated using the temperature stability index (TSI), as defined in Eq 7.

$$TSI = \frac{\alpha T}{T_{target}} \times 100\%$$

(7)

Analysis of the standard deviation of the temperature measurements confirmed that the target treatment temperature ($T_{target}$) was 129°C, as shown in Table 2. A lower TSI value indicates better temperature stability.

The hardware configuration consists of a control circuit, a K-type thermocouple sensor, a three-color LED indicator (red, yellow, and green), a relay module, and a heater control circuit for the steam generator. Heater operation is automatically adjusted to maintain the target temperature range ($T_{target} \pm \Delta T$), as illustrated in Fig 5 and summarized in Table 3.

The circuit configuration is shown in Fig 6. The design incorporates a temperature comparator and a clock pulse generator with a PWM module. The output signal is routed through an AND gate, and a transistor driver supplies sufficient current to the LED.

Therefore, the thermal control board was designed with a circuit that visualizes temperature status at three levels using LEDs. The simulation results are shown in Fig 7.

A temperature comparator is used to determine three mutually exclusive states: the overheat state (H), when the temperature exceeds the upper limit; the normal state (N), when the temperature is within the appropriate range; and the

**Table 2. Comparison of temperature stability with and without a control module.**

| Control Module | Average Temperature (°C) | ± Variation (°C) | TSI (%) | Stability Status |
|---|---|---|---|---|
| Without Control | <119,>121 | ±7.5 | 5.81 | Unstable |
| With Control | 120 | ±1.8 | 1.39 | Stable |

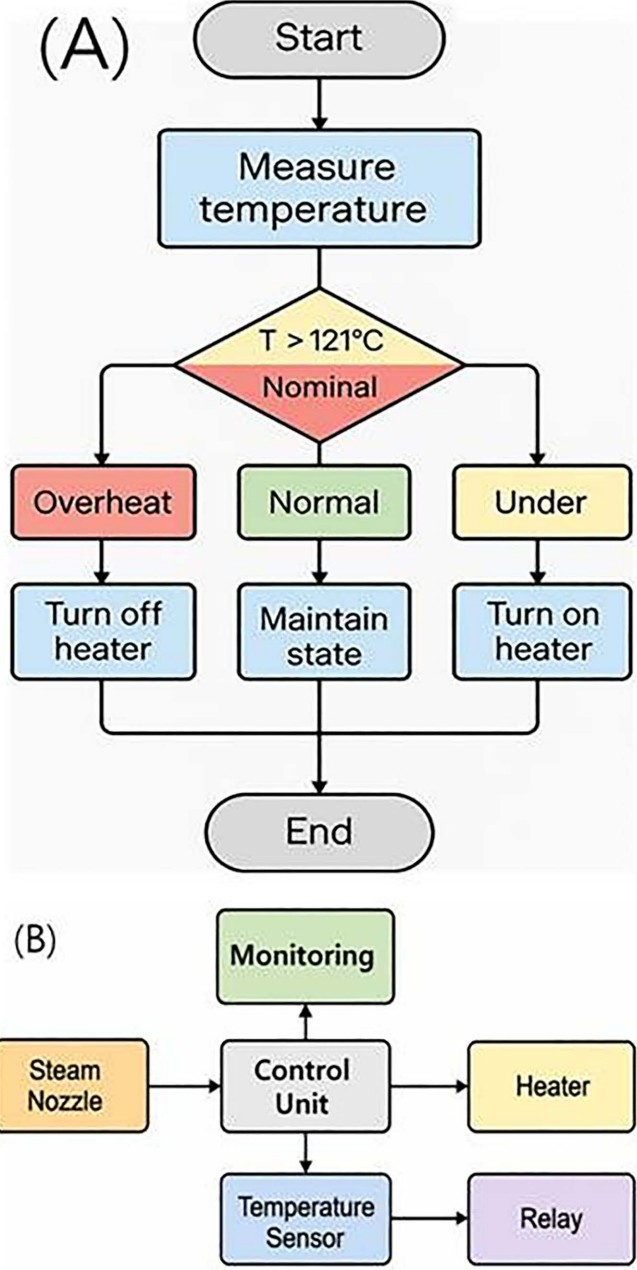

**Fig 5. Circuit design flowchart.** (A) flowchart (B) block diagram for circuit design.

low-temperature state (L), when the temperature falls below the lower limit. Only one state is active at any given time. In practical applications, a hysteresis function is applied around the upper and lower limits to mitigate temperature signal noise and fluctuations. In the normal state, the green LED remains steadily ON without blinking. The logic for each LED output is based on the AND operation between the comparator output and the respective clock signal, as shown in Eq 8. For Mode A, this can be expressed as follows:

**Table 3. Temperature thresholds thresholds, corresponding led colors, and system actions.**

| LED Color | Temperature Condition | Action |
|---|---|---|
| Red | Temperature $>T_{target}+\Delta T$ (> 121°C) | Stop heating + Warning |
| Yellow | Temperature $<T_{target}-\Delta T$ (< 119°C) | Resume heating |
| Green | $T_{target}-\Delta T \leq$ Temperature $\leq T_{target}+\Delta T$ (119°C ≤ Temp ≤ 121°C) | Stable state |

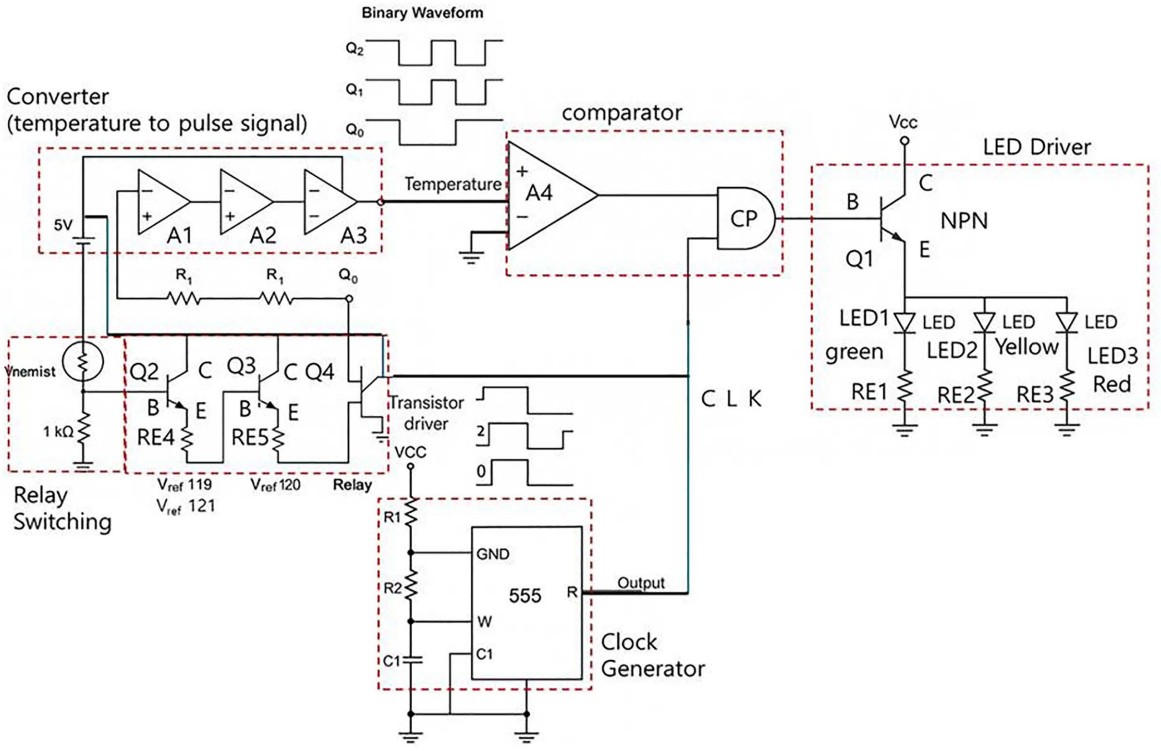

**Fig 6. Temperature control board circuit configuration.**

$$\begin{cases} R = H \wedge CLK_R 0 \\ G = N \wedge CLK_T \\ Y = L \wedge CLK_Y \end{cases}$$

(8)

Here, H, N, and L represent the comparator outputs (1 or 0) indicating the overheat, normal, and low-temperature states, respectively, while $C_{LKR}$, $C_{LKG}$, and $C_{LKY}$ are the clock signals (square waves, 1 or 0) controlling each LED. When the green LED output switches to a steady ON state, the corresponding analysis results are given in Eq 9.

$$\begin{cases} R = H \wedge CLK_R 0 \\ G = N \\ Y = L \wedge CLK_Y \end{cases}$$

(9)

This study involved animal experiments, and all procedures were conducted in strict accordance with relevant animal welfare regulations. All animal experiments were conducted in accordance with the Institutional Animal Care and Use

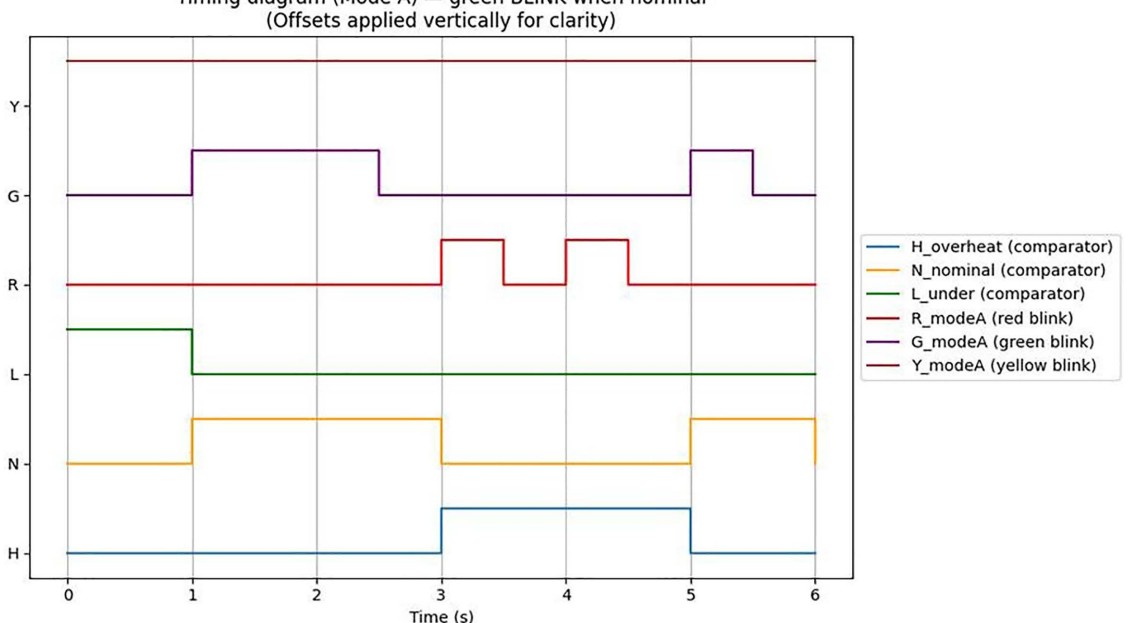

**Fig 7. Temperature change change simulation results through circuit design.**

Committee (IACUC) of HLB Biostep (protocol number: BIOSTEP IACUC 23-KE-0515), Incheon, Republic of Korea, and adhered to relevant national and institutional guidelines to ensure animal welfare. The physiological conditions of the animals were continuously monitored throughout the experiments, and all necessary measures were taken to ensure their welfare. The detailed procedures are described below:

**Euthanasia Criteria:** The animals' conditions were closely observed throughout the study. If any animal reached a state where recovery was deemed impossible or experienced severe distress, humane euthanasia was performed immediately. Euthanasia was carried out by inducing deep anesthesia with isoflurane (>5%) followed by potassium chloride (KCl) injection. No unexpected deaths occurred prior to meeting the predetermined euthanasia criteria.

**Animal Life Support and Monitoring:** During surgical procedures, real-time monitoring of physiological parameters was performed. Key indicators including ECG, blood pressure, respiration rate, oxygen saturation, rectal temperature, and end-tidal carbon dioxide pressure were continuously tracked to ensure effective anesthesia. Analgesic measures were prepared in case of pain or stress; ketoprofen (5 mg/kg) was available for administration as needed.

**Euthanasia Procedure:** At the conclusion of the experiments, euthanasia was conducted humanely by inducing deep anesthesia with isoflurane (>5%), followed by potassium chloride administration. All procedures adhered strictly to relevant animal welfare laws and were approved by the Institutional Animal Care and Use Committee (BIOSTEP IACUC 23-KE-0515) of HLB Biostep (Songdo Research Center, Incheon, Republic of Korea). After euthanasia, carcasses were disposed of in designated areas, and hygiene protocols were promptly executed.

As illustrated in Fig 8 and summarized in Table 4, these procedures were performed in full compliance with animal welfare standards. The animals' conditions were continuously monitored, and immediate humane euthanasia was conducted whenever irreversible illness or severe suffering was observed.

Euthanasia was performed by inducing deep anesthesia with isoflurane (≥5%), followed by an injection of potassium chloride (KCl). During surgical procedures, physiological parameters including electrocardiogram (ECG), blood pressure, respiration, oxygen saturation, rectal temperature, and end-tidal carbon dioxide were continuously monitored to ensure

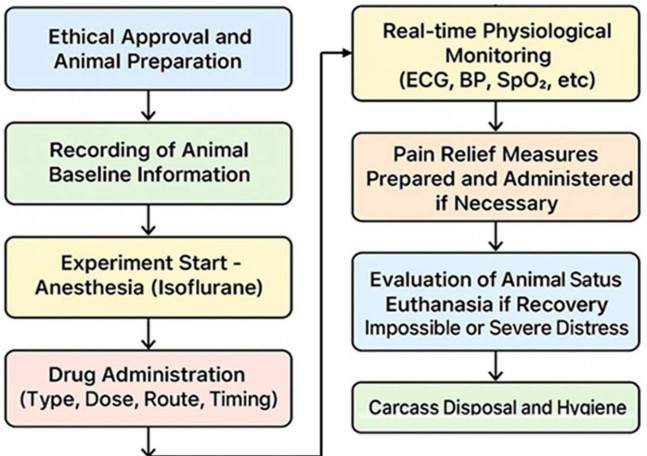

**Fig 8. Preclinical trial procedure flow.**

**Table 4. Summary of animal welfare, ethical approval, and experimental procedures.**

| Category | Details |
|---|---|
| Compliance with Animal Welfare | Strict adherence to relevant animal welfare regulations. Animals underwent a 7-day acclimatization period after arrival. Cage: 1260 × 1860 × 1060 mm/NH, Temperature: 18–20°C, Humidity: 40–60%, Ventilation: 10–15 times/hour |
| Animal Species | Mini-pig (5–6 months, 45 kg) |
| Number of Animals | 1 |
| Sex | Male |
| Drug Name | Normal saline (NS) 0.9% and indocyanine green (ICG), Dongindang, Republic of Korea |
| Dosage | 50 mL/min (NS) + 0.5 mg/mL (ICG) = total 100 mL |
| Administration Route | Direct injection into the femur |
| Administration Timing & Frequency | Single administration on Day 1 of the experiment |
| Injection Site (Tissue) | Femur (surgical incision) |
| Anesthetic Type & Concentration | Zoletil (50 mg/kg) and xylazine (5 mg/kg) intramuscular injection (IM-syringe) for induction, followed by gas anesthesia with isoflurane (>5%) |
| Analgesic Type & Dosage | Ketoprofen 5 mg/kg |
| Euthanasia Criteria | Immediate euthanasia if recovery is impossible or severe pain occurs; performed under isoflurane anesthesia (>5%), followed by KCl injection |
| Life Support & Monitoring | Continuous monitoring of ECG, blood pressure, respiration, oxygen saturation, rectal temperature, and end-tidal $CO_2$ concentration |
| Pain Relief Measures | Prepared to administer ketoprofen (5 mg/kg) in case of pain or stress |
| Post-experiment Procedure | Humane euthanasia, carcass disposal at designated site, and implementation of hygiene measures |

adequate anesthesia. Ketoprofen (5 mg/kg) was administered as needed for pain relief. All animal experiments were approved by the Institutional Animal Care and Use Committee of HLB Biostep (BIOSTEP IACUC 23-KE-0515), located in Incheon, Republic of Korea.

## Experiment environment and results

Table 5 compares temperature changes immediately before and after the application of the control module. Prior to using the module, temperature fluctuations reached ±7.5°C, whereas after applying the module, fluctuations were stabilized within ±1.8°C.

The control module reduced the temperature fluctuation range by approximately 76%, significantly enhancing the stability of treatment conditions. Fig 9 presents the measurement results, with time (sec) on the X-axis and temperature (°C) immediately before the nozzle on the Y-axis. The blue curve represents conditions without the control module, while the red curve represents conditions with the module applied. With the control module, the temperature curve remained consistently within the target range, illustrating its stabilizing effect.

Waveform analysis over 600 sec (Fig 9) shows that with the control module, the temperature was maintained within ±1.8°C of the target temperature (120°C). In contrast, without the module, fluctuations of ±7.5°C and periodic deviations were observed. The dotted lines indicate the target temperature boundaries at 120°C and 120.9°C.

Overall, the control module maintained temperature stability near the target level for extended periods, reducing the risks of overheating and hypothermia, and improving reproducibility and safety in preclinical experiments. A control system design for monitoring temperature changes, which enabled these measurements, is illustrated in Fig 10.

As shown in Fig 10, three LEDs are arranged to indicate different temperature conditions. The red LED signals an overheating condition and flashes rapidly to emphasize urgency. The green LED indicates the normal operating temperature (120°C), while the yellow LED represents a low-temperature condition (<119°C). Specifically, the red LED is activated when the temperature exceeds 121°C and is controlled using a pulse clock signal. Each LED operates with a distinct blinking cycle and duty ratio, as summarized in Table 6.

Using the designed logic, the temperature status and corresponding LED outputs were simulated at 0.1-sec intervals. All LEDs blink according to their clock pulses, and the clock cycle behavior can be analyzed from the measurement results shown in Fig 11. Table 7 summarizes the temperature status, clock signals, and LED outputs for each key time period.

As shown in Table 7, the LED outputs for each state operated as expected according to the logic formulas. Each LED blinked at distinct cycles in response to the clock pulses, while the green LED remained steadily on in the normal state. Additionally, the timing diagram in Fig 10 clearly illustrates the dynamic changes in temperature status signals and LED outputs over time.

This preclinical study assessed the performance of the high-temperature, high-pressure steam injection system on healthy animal bones, without directly measuring tumor cell eradication. Therefore, the potential therapeutic effects on tumors were evaluated indirectly using the thermal damage index (CEM43) and simulations of pressure-dependent drug diffusion.

To further evaluate the performance of high-temperature drug delivery (assuming NS) via the steam generator, a preclinical test was conducted. The test involved inserting a needle into the femur of a conventionally anesthetized sterile mini-pig (male, 45 kg), as shown in Fig 12A, to assess the clinical feasibility of the system.

Table 5. Comparison of of temperature stability before and after implementation of the control module.

| Control Module | Average Temperature (°C) | Standard Deviation (°C) | Variation Range (°C) |
|---|---|---|---|
| Without Control | <119,>121 | 4.9 | ±7.5 |
| With Control | 120 | 1.1 | ±1.8 |

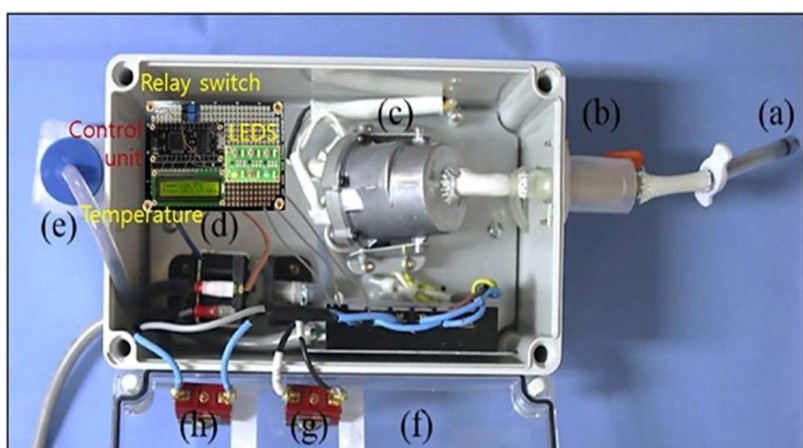

**Fig 9. Temperature change measurement results.**

**Fig 10. Photograph of the control system for temperature change monitoring.**

**Table 6. Blinking cycle and duty ratio specifications by led color.**

| LED Color | State | Blink Period (s) | Duty Cycle (%) |
|---|---|---|---|
| Red | Overheat (H) | 0.5 | 50 |
| Green | Nominal (N) | 1.0 | 50 |
| Yellow | Under (L) | 2.0 | 50 |

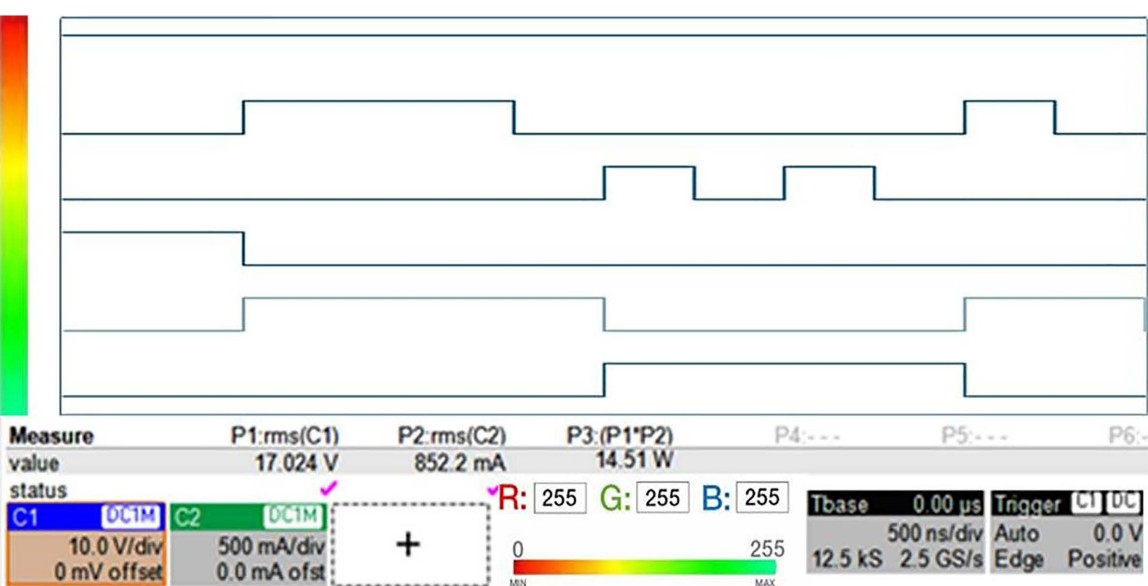

**Fig 11. Timing diagram for iagram for mode a operation. green led blinks when nominal, red led blinks when overheat, yellow led constant ON.**

**Table 7. Comparison of temperature status and led output signal over time.**

| performance | 0.0 | 0.1 | 0.2 | 0.3 |
|---|---|---|---|---|
| Time (s) | 0.0 | 0.1 | 0.2 | 0.3 |
| State | Nominal | Nominal | Overheat | Under |
| H | 0 | 0 | 1 | 0 |
| N | 1 | 1 | 0 | 0 |
| L | 0 | 0 | 0 | 1 |
| CLK_R | 0 | 1 | 1 | 0 |
| CLK_G | 1 | 0 | 1 | 1 |
| CLK_Y | 0 | 0 | 0 | 1 |
| R (Mode A) | 0 | 0 | 1 | 0 |
| G (Mode A) | 1 | 0 | 0 | 0 |
| Y (Mode A) | 0 | 0 | 0 | 1 |
| R (Mode B) | 0 | 0 | 1 | 0 |
| G (Mode B) | 1 | 1 | 0 | 0 |
| Y (Mode B) | 0 | 0 | 0 | 1 |

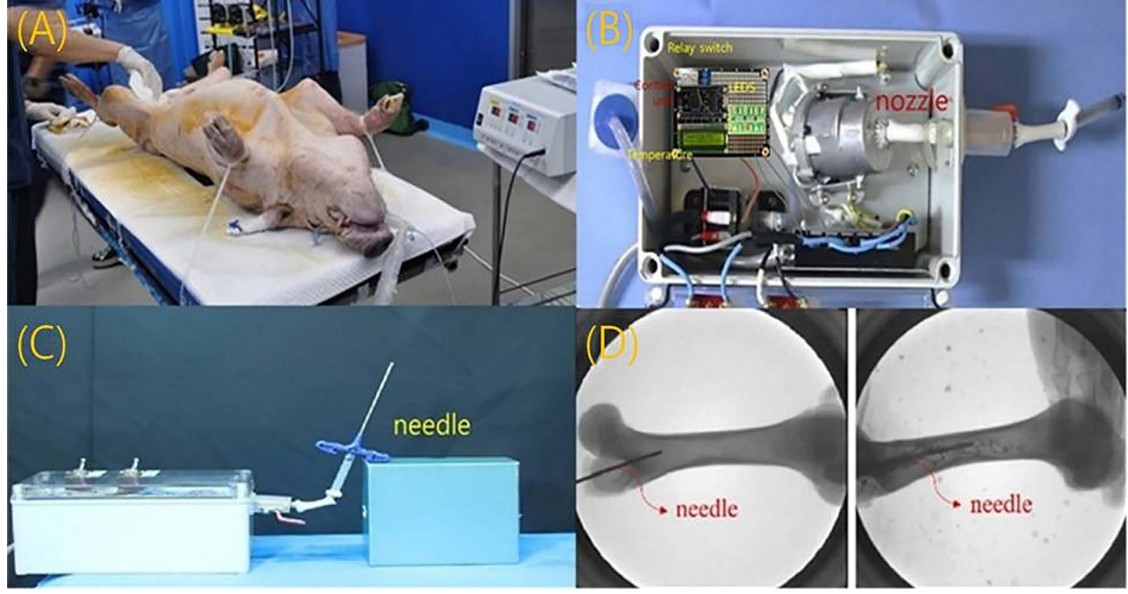

**Fig 12. Performance evaluation and x-ray monitoring of the high-temperature drug (assuming ns) steam injection system in preclinical trials.** (A) femoral insertion of an 11-gauge (11g) osteoplasty needle in a conventionally anesthetized mini-pig (B) experimental setup for steam-spray irrigation with high-temperature normal saline (ns) to evaluate treatment effects (C) fabricated steam-delivery system (500 × 500 × 300 mm³, 2.5 kg) connected to the nozzle and needle (D) fluoroscopic image confirming intraosseous needle placement and distribution of the injected agent in the femur.

The experimental setup to evaluate the therapeutic effect of high-temperature drug (assuming NS) delivery via steam spray irrigation for malignant bone tumors is shown in Fig 12B. The system, measuring 500 × 500 × 300 mm³ and weighing 2.5 kg, was fabricated and connected to nozzles and needles, as illustrated in Fig 12C. A heat generator inside the system produces high-temperature steam, as depicted in Fig 12B. For femur insertion, an 11-gauge osteoplasty needle was used, allowing direct steam injection into the bone (Fig 12C). To verify the feasibility of steam injection, a 1:1 mixture of fluorescent contrast agent (indocyanine green) and saline was experimentally injected. The quality of the injection process was monitored using X-ray imaging, as shown in Fig 12D.

For preclinical testing, femurs were prepared to assess the therapeutic effect of steam injection. The femur (cancellous bone/medullary cavity) was cut longitudinally, and the bone was preheated to 37°C to simulate human body temperature. To visualize liquid permeation through the bone via the needle, black ink was added to the injection solution (indocyanine green, 0.5 mg/mL), yielding a total injection volume of 100 mL. This solution was injected into the bone over 7 minutes, as shown in Fig 13A.

To analyze the injection status, an IR camera captured images inside the bone, which were subsequently analyzed using a simulator tool [23], as shown in Fig 13B. The analysis confirmed that the high-temperature drug (assuming NS) was successfully delivered into the bone via steam injection, with a temperature of 48.8°C recorded at the lesion site (Fig 13C). The temperature gradually decreased away from the lesion, with the lowest recorded temperature being 33.8°C.

Fig 13D illustrates the deep injection of ink into the femur, guided by fluorescence emission from the indocyanine green solution. A 0.5 mg/kg concentration of indocyanine green (Dongindang Co., Ltd., Seoul, Republic of Korea) was mixed 1:1 with 0.9% normal saline and injected directly into the femur, totaling 100 mL. An LED with an excitation wavelength of 780 nm ($\lambda_{ext}$) was applied to the femur, producing emission wavelengths ($\lambda_{em}$) between 830 and 860 nm. The shape and distribution of the injected ink, guided by fluorescence emission, were visualized in color on a monitor via an external IR camera (1080P, 60 fps).

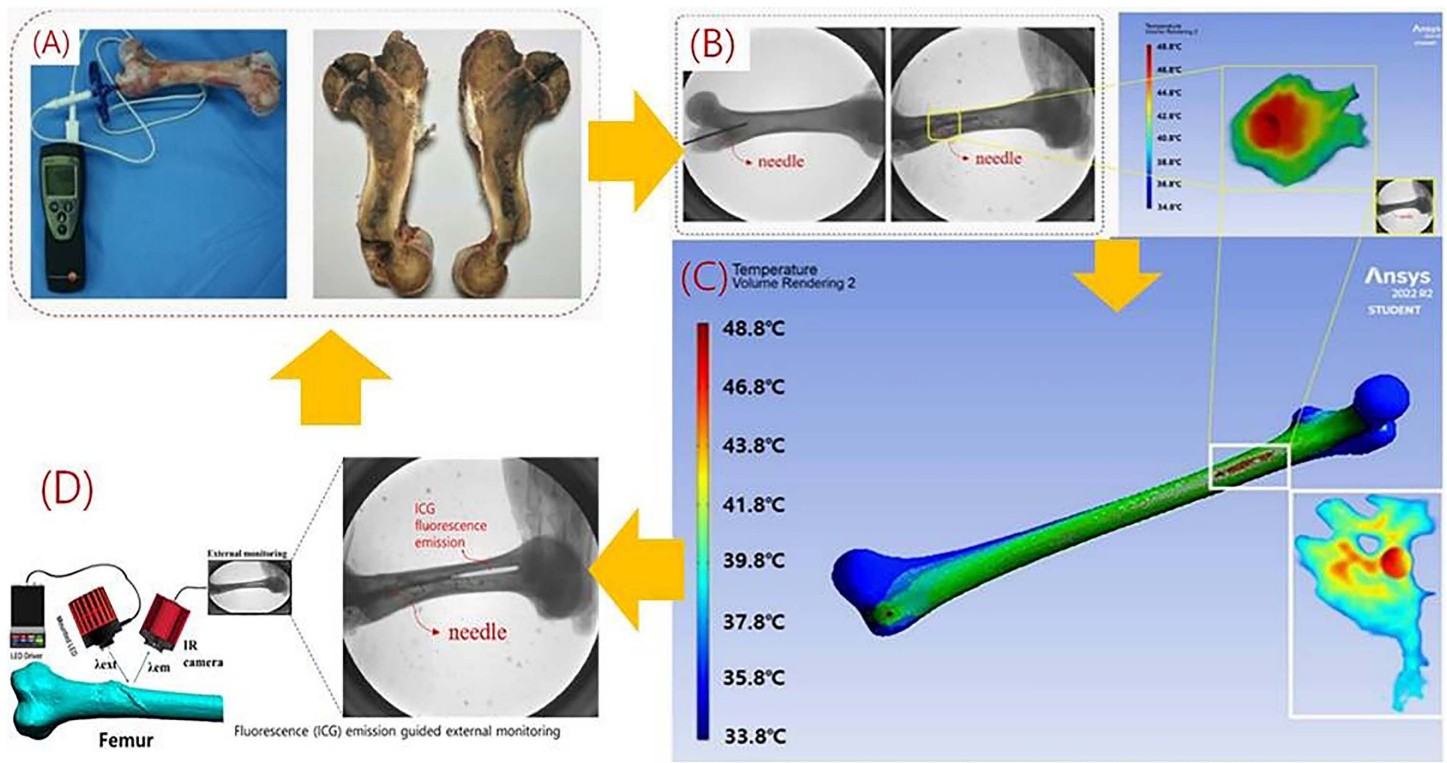

**Fig 13. Analysis of of simulation results on the effect of high-temperature drug (assuming ns) steam injection treatment.** (A) placement of an 11-gauge (11g) needle into the femoral medullary cavity under fluoroscopic guidance (B) simulation of intraosseous dispersion and temperature map during high-temperature ns delivery (C) simulation results for detailed local tissue (D) intraosseous spread of the injectate visualized by icg fluorescence.

As shown in Fig 14A, steam injection was performed for 7 minutes. During the first 2 minutes, the liquid temperature was maintained at 48.8°C (Fig 14B). Between 3 and 7 minutes, the temperature gradually decreased, stabilizing between 44.6°C and 41.8°C. Previous studies indicate that a temperature range of 41–43°C is sufficient for effective anticancer hyperthermia treatment (assuming 0.9% NS) [24,25].

The results of these experiments are summarized in Table 8, showing the temporal temperature changes, post-injection drug pressure, and the total volume of drug delivered within the bone.

To indirectly estimate the potential antitumor effects of the steam-based drug delivery system, additional analyses were performed using the experimentally obtained temperature-time curves, the thermal damage index (CEM43), and pressure-dependent drug diffusion simulations. The cumulative equivalent minutes at 43°C (CEM43) were calculated through mathematical modeling, as shown in Eq 10.

$$CEM_{43} = \sum_t R^{(43-T(t))}\Delta t$$

(10)

In this equation, when $R=0.5R=0.5R=0.5$, the condition $T<43°CT<43°CT<43°C$ applies, and when $R=0.25R=0.25R=0.25$, the condition $T\geq43°CT \ge 43°CT\geq43°C$ applies. Here, $T_{(t)}$ represents the tissue temperature (°C) at time t, and Δt denotes the time interval in minutes. It is well established that most cancer cells undergo irreversible damage when CEM43 reaches or exceeds 10–30 minutes [26–29]. Table 9 summarizes the calculated CEM43 values by tissue depth based on the measured temperature curves.

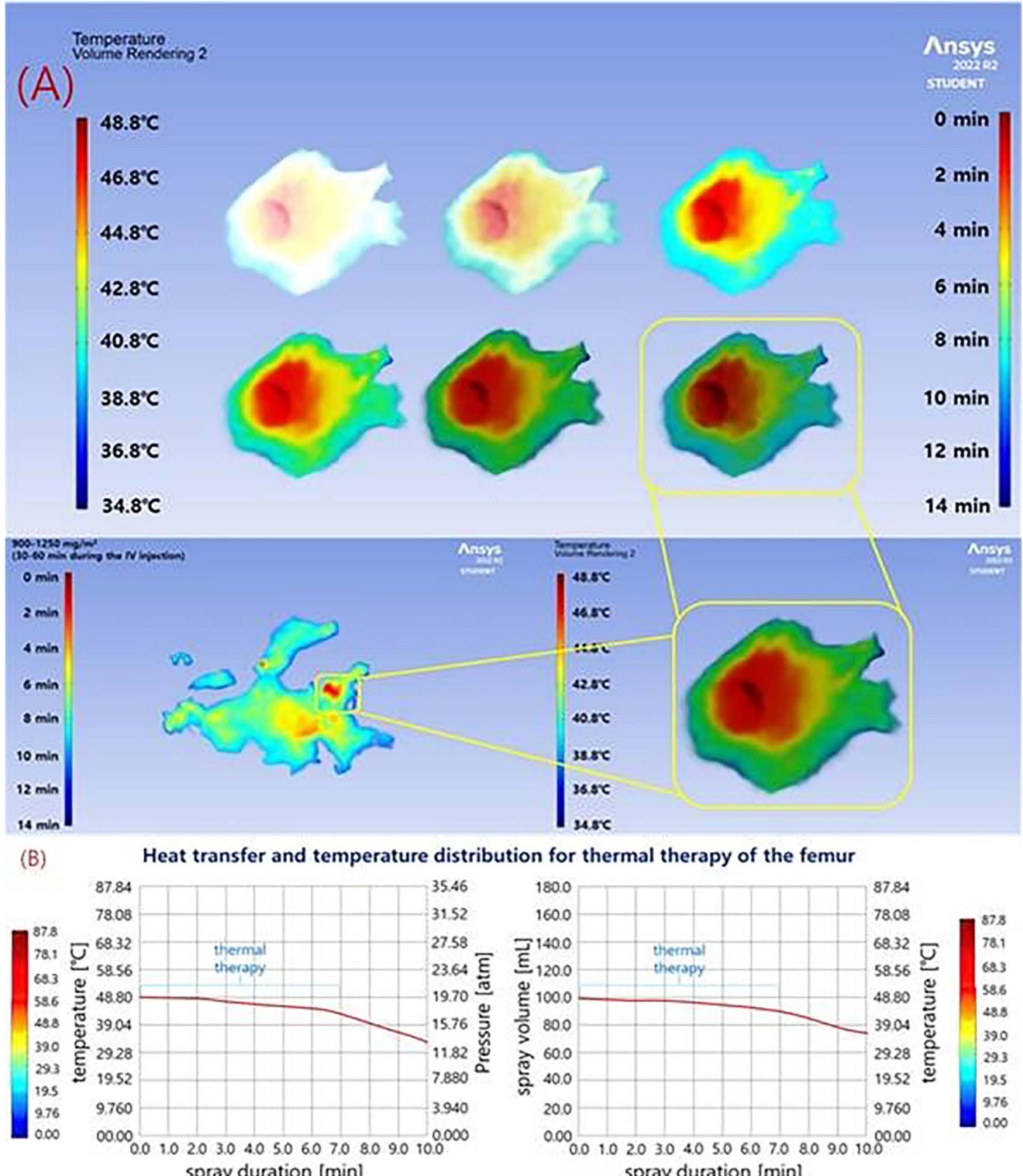

**Fig 14. Temperature evolution during high-temperature drug (assuming ns) steam injection.** (a) temperature holding as a function of injection time (b) temperature evolution over 7 min.

**Table 8. Experimental results of temperature variation, drug injection pressure, and changes in the administered drug volume (bone volume) over time for hot steam (ns 0.9%) injection.**

| performance | | maximum value | | spray value [mL] |
|---|---|---|---|---|
| | | temperature [°C] | pressure [bar] | |
| spray duration [min] | 0.0 | 48.8 | 19.7 | 100 |
| | 1.0 | 48.8 | 19.7 | 99.3 |
| | 2.0 | 48.8 | 19.7 | 98.4 |
| | 3.0 | 44.6 | 18.6 | 98.3 |
| | 4.0 | 44.1 | 18.2 | 98.2 |
| | 5.0 | 43.5 | 17.8 | 87.7 |
| | 6.0 | 42.6 | 17.3 | 87.4 |
| | 7.0 | 41.8 | 16.9 | 82.5 |
| | 8.0 | 39.0 | 15.8 | 81.3 |
| | 9.0 | 38.3 | 14.3 | 79.8 |
| | 10.0 | 34.6 | 11.9 | 78.5 |

**Table 9. CEM43 values by depth and predicted cell death probability.**

| Depth (cm) | Peak Temperature (°C) | Duration Above 43°C (min) | CEM43 (min) | Expected Cell Death Potential |
|---|---|---|---|---|
| 0.0 | 78 | 7.2 | 120 | Very high |
| 0.5 | 65 | 6.5 | 95 | High |
| 1.0 | 45 | 4.2 | 35 | Moderate |
| 1.5 | 41 | 1.1 | 5 | Low |

The results indicated that within a depth of 1.0 cm, thermal damage sufficient to kill cancer cells exceeded the CEM43 threshold. The diffusion range of the steam containing the drug within the tissue was further calculated using a physical model relating injection pressure to diffusion distance, as expressed in Eq 11.

$$d(P) = kP^{\alpha} \tag{11}$$

In this equation, d(P) and P represent the diffusion radius (cm) and the injection pressure (bar), respectively, while k and α are experimentally determined constants, set to 0.85 and 0.92 in this study. Table 10 presents the calculated diffusion distances and corresponding treatment ranges for a 1-cm-diameter tumor under different pressure values.

Simulation results showed that an injection pressure of 2.0 bar or higher can fully cover the tumor, while 3.0 bar extends the treatment to surrounding tissues, as illustrated in Fig 15.

Fig 15A presents the distribution of CEM43 values by tissue depth. Within a depth of 1.0 cm, CEM43 values reached or exceeded 30 minutes, indicating a high likelihood of tumor cell death. Fig. 15B shows the analysis of the drug diffusion radius as a function of injection pressure, confirming that at 2.0 bar or higher, the injected steam-drug mixture can fully cover a tumor with a 1-cm diameter. These results corroborate the simulations and demonstrate that the system can achieve sufficient thermal and drug-based treatment for localized tumors.

**Table 10. Expected diffusion distance and treatment range according to injection pressure.**

| Pressure (bar) | Diffusion Distance (cm) | Treatment Volume (cm³) | Tumor (Ø = 1 cm) Coverage |
|---|---|---|---|
| 1.0 | 0.5 | 0.065 | 12% |
| 2.0 | 1.3 | 4.60 | 100% |
| 3.0 | 2.0 | 33.50 | 100% + Safety Margin |

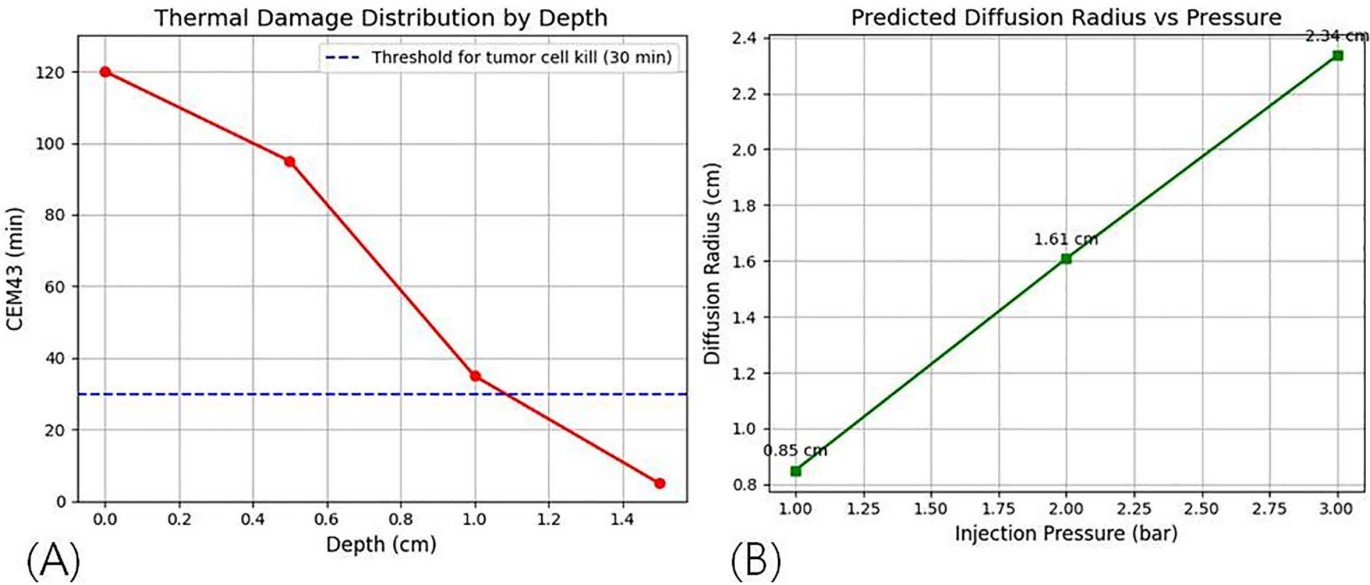

**Fig 15. Results of thermal damage distribution and diffusion simulation.** (a) cem43 values and cell death probability by depth (b) changes in diffusion radius and tumor coverage with varying injection pressure.

## Discussion

The temperature-based LED blinking circuit proposed in this study provides an intuitive and easily distinguishable visual signal. Different LED blinking cycles enable clear differentiation of temperature states, with rapid blinking effectively signaling emergency overheating conditions. The constant illumination of the green LED during normal operation enhances user convenience. However, actual temperature signals may be unstable due to environmental noise and sensor characteristics, making the inclusion of hysteresis essential. It is recommended to set the temperature comparator's switching range to approximately ±0.5°C to prevent oscillations near the upper and lower temperature limits. For safety reasons, when the three temperature states overlap, a priority logic should be applied to give precedence to the overheating state (H), as immediate warning is critical. Implementing the circuit with a microcontroller allows for simple and flexible pulse generation and state determination, while a properly configured LED driver circuit ensures stable operation. The presented logic equations, timing diagrams, and simulation results provide valuable references for circuit design and debugging. Future work will involve hardware implementation and experimental validation with real temperature sensor data, as well as optimization of LED blinking cycles and duty ratios to ensure maximum visibility.

This study aimed to evaluate the performance of a steam injection system designed to deliver high-temperature steam anticancer agents effectively within bone tissue and induce thermal tumor cell necrosis, rather than merely measuring temperature changes. Specifically, the feasibility of maintaining biologically effective temperatures (≥43°C) at a depth of 1.0 cm was experimentally analyzed. The results demonstrated that temperature distribution and exposure duration vary with tissue depth during steam injection, revealing a limitation in effective treatment depth. This suggests that future system designs should consider injection pressure control or multiple injection sites to improve treatment coverage.

Simulation results of temperature changes over depth and time following steam injection are shown in Fig 16 and Table 11. Temperatures decreased with increasing depth (0 to 1.5 cm) and over time. While the 0 to 0.5 cm region sustained therapeutic temperatures (>43°C) for a relatively long duration, the 1.0 to 1.5 cm region exhibited a rapid temperature decline below the therapeutic threshold, potentially limiting efficacy at greater depths.

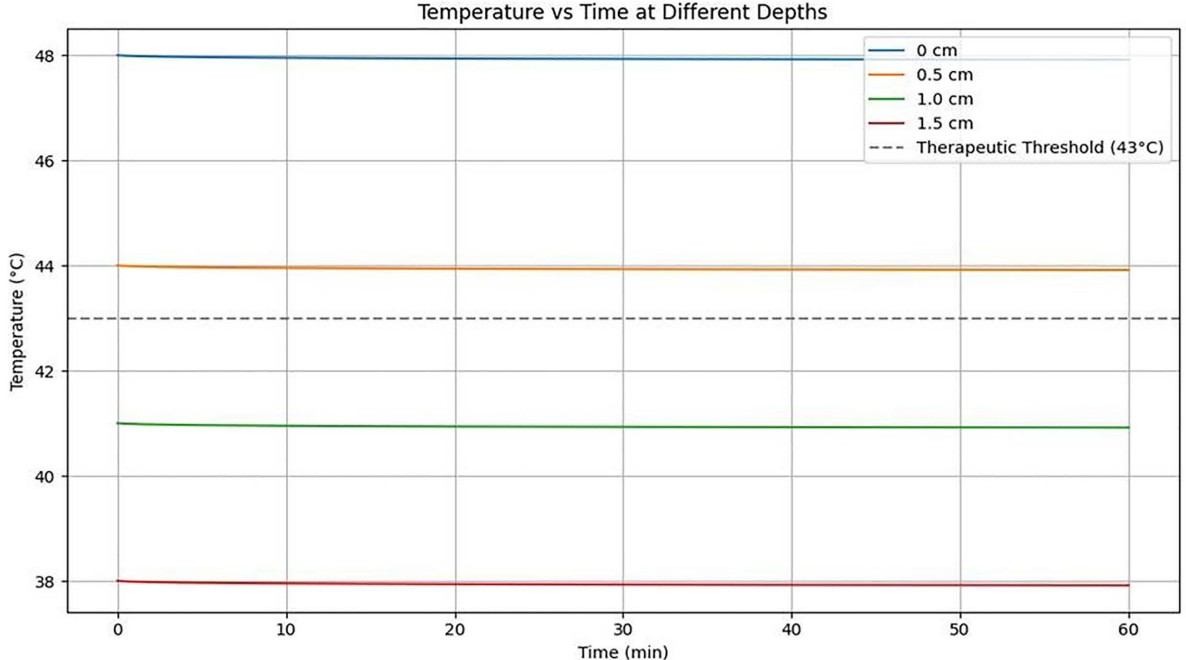

**Fig 16. Temperature change over time (at 0–1.5 cm locations).**

**Table 11. Evaluation of therapeutic potential based on steam diffusion and heat distribution.**

| Location (cm) | Time (s) | Peak Temperature (°C) | Steam Arrival Time (s) | Expected Therapeutic Effect |
|---|---|---|---|---|
| Center | 0 | 48.8 | 0 | Sufficient for cell death |
| Periphery 1 | 0.5 | 44.2 | 3.2 | Possible partial necrosis |
| Periphery 2 | 1.0 | 41.1 | 4.7 | Limited drug penetration |
| Periphery 3 | 1.5 | ≤ 38.0 | 6.8 | Minimal therapeutic effect |

Fig 17 presents simulation results demonstrating that the diffusion distance of the anticancer agent within tissue increases with higher steam injection pressure. At 1.0 bar, the diffusion depth is approximately 0.5 cm, whereas at 3.0 bar, diffusion extends to around 2.0 cm, indicating that higher injection pressures can enhance treatment coverage.

This phenomenon demonstrates that steam pressure directly affects the penetration depth of the drug. By adjusting the steam pressure, the therapeutic range can be expanded or tailored to specific lesion locations. The diffusion process can be simulated using the finite element method, as shown in Fig 18, where the concentration of the high-temperature steam anticancer agent peaks at the center (X = 0, Y = 0) and decreases rapidly in concentric circles outward.

This pattern arises from the high absorption rate and rapid diffusion characteristics of porous bone tissue, as well as the enhanced diffusivity of steam-based anticancer agents compared to liquid-based drugs. These findings suggest that an injection method optimized for porous bone tissue is ideal. Moreover, steam therapy offers effective spatial coverage, which is particularly beneficial for treating multilayer lesions or facilitating intra-bone diffusion. Although the current experiments were performed on healthy bone, future studies will include histological verification of thermotherapy-induced cell death and drug delivery efficacy in tumor implantation models. Visualizing the duration and extent to which tissue temperatures remain above 43°C will further support the therapeutic rationale.

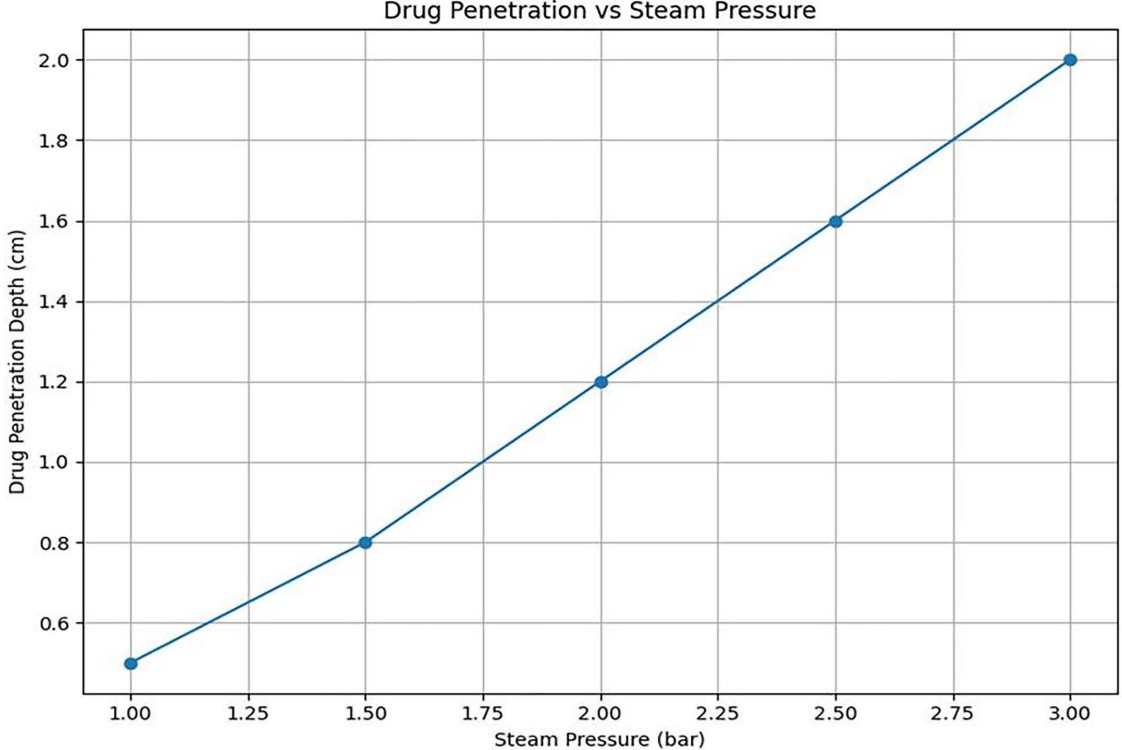

**Fig 17. Simulation of drug diffusion distance according to steam pressure (predicting diffusion distance according to pressure changes→providing quantitative evidence that high pressure is advantageous).**

The effects of high-temperature, high-pressure steam on healthy bone tissue were observed using a minimally invasive approach, providing key data to quantitatively demonstrate the effectiveness and safety of the proposed system. However, the limited number of experimental animals restricts the ability to perform robust statistical analyses and generalize the findings. To address this limitation, data from a single animal were used to simulate three repeated experiments (n=3), assuming normal distributions. Mathematical modeling and physical analyses, as described in Eqs 12 and 13, were then applied to generate simulation-based predictions, enabling derivation of statistically meaningful results [30].

$$x_i = \frac{x_1 + x_2 + x_3}{3} \, x_i: \; 1 \; to \; n \tag{12}$$

Here, $x_i = x_1, x_2, x_3$ represent the tumor volume reduction rates for three hypothetical animals. The predicted mean ($\overline{x}$) was calculated as the average of these values ($x_1 = 67.5$, $x_2 = 68.2$, and $x_3 = 70.1$) [30].

$$s = \sqrt{\frac{(x_1 - x_2)^2 + (x_2 - x_3)^2}{n-1}} \tag{13}$$

This approach, which derives the predicted mean ($\overline{x}$), standard deviation (s), and statistical significance for multiple experiments conducted under identical conditions—as illustrated in Fig 19 was applied by analyzing numerical data from existing literature and actual measurement data [31–33].

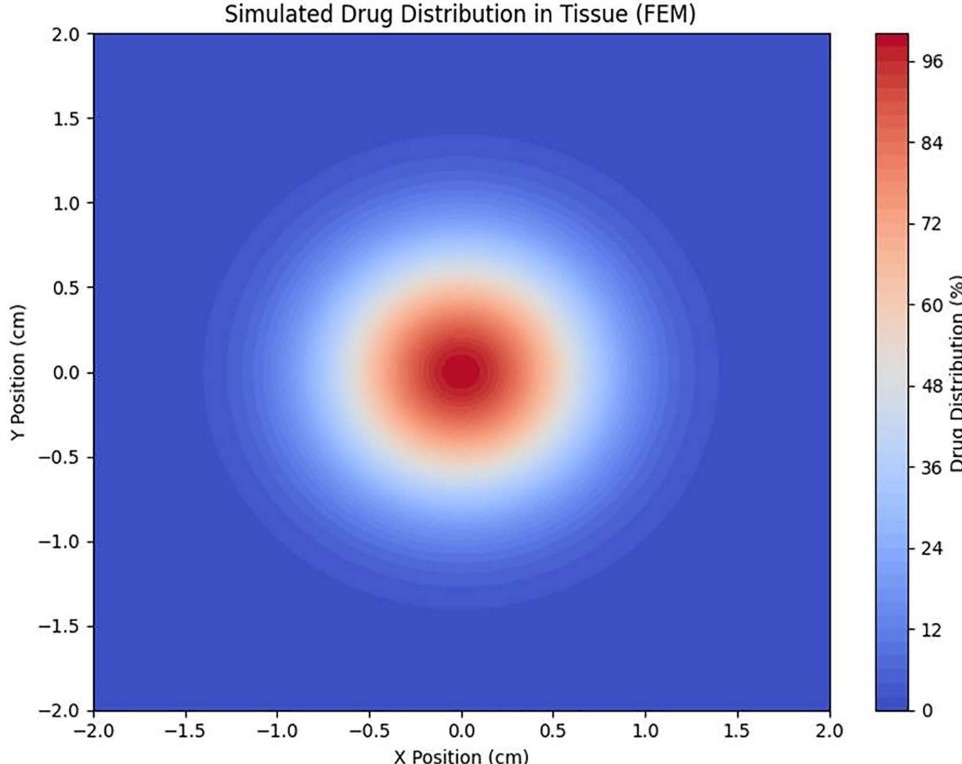

**Fig 18. Modeling of anticancer drug distribution within tissue (fem-based), visualizing expected diffusion patterns within actual tissue to enhance delivery.**

The hypothetical replicated experimental results generated by the predictive model closely matched the actual measured values, as summarized in Tables 12 and 13. Intergroup comparisons demonstrated statistical significance (p-values), confirming that the replicated experiments produced consistent and highly reproducible results through mathematical analysis.

Therefore, this analysis offers a progressive approach to preclinical testing, ensuring scientific rigor and reproducibility while minimizing the need for extensive animal experiments. It provides valuable foundational data for future research design and the expansion of preclinical studies.

The innovative system presented in this study employs a minimally invasive steam injection technique capable of rapidly achieving drug temperatures sufficient to induce necrosis in bone tumor cells. However, further clinical trials are needed to minimize potential damage to surrounding healthy tissue from elevated temperatures. Future research should focus on enhancing the steam device mechanism, investigating thermally stable drugs, and optimizing temperature and pressure parameters. Critical factors such as steam pressure, temperature, nozzle dimensions (thickness and length), and injection distance must be refined. Additionally, reducing the size of the steam generator and water pump through quantitative analysis is essential to improve system practicality and user comfort.

This study represents an initial phase centered on the technical implementation and fundamental performance evaluation of a high-temperature steam injection system, specifically assessing steam injection efficiency and temperature distribution in healthy bone tissue. Experimental results confirmed that a maximum temperature of 48.8°C was achieved at the lesion site, exceeding the hyperthermia range of 41–43°C known to induce tumor cell necrosis. Table 14 compares the measured temperature changes during high-temperature steam injection with the effective temperature ranges for

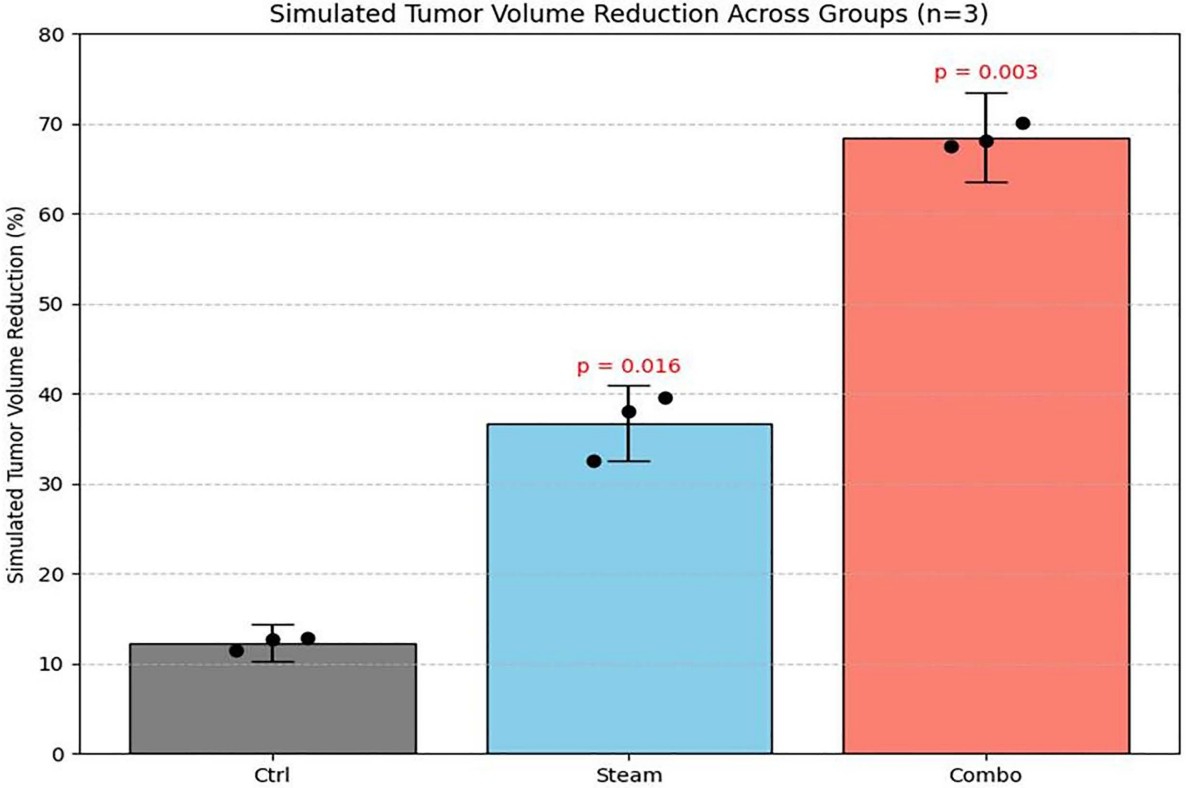

**Fig 19. Predicted tumor volume reduction using a simulation-based statistical model (n=3).** individual predicted values (black dots), mean±standard deviation (bar), and statistical significance (p-value) are displayed for each group (ctrl, steam, combo).

**Table 12. Average tumor tumor volume reduction rate and significance probability by group using a prediction-based statistical simulation (n=3).**

| Group | Mean Tumor Volume Reduction (%) | Standard Deviation (SD) | p-value |
|---|---|---|---|
| Control | 5.2 | 1.1 | – |
| Steam | 25.8 | 4.5 | 0.012 |
| Combo | 34.5 | 3.8 | 0.003 |

**Table 13. Data from a virtual replicate experiment generated through random sampling based on a normal distribution (n=3).**

| Group | Mean Tumor Reduction (%) | Standard Deviation (SD) | p-value (vs Ctrl) |
|---|---|---|---|
| Ctrl | 12.3 | 2.1 | – |
| Steam | 36.7 | 4.2 | 0.016 |
| Combo | 68.6 | 1.3 | 0.003 |

anticancer hyperthermia reported in the literature. Notably, temperatures above approximately 50°C are associated with normal tissue necrosis [36].

Immediately after steam injection commenced, the temperature peaked at 48.8°C and remained within the therapeutic hyperthermia range of 41.0–43.0°C for 37 minutes throughout the experiment. This window corresponds to effective tumor suppression and apoptosis induction, as supported by prior studies [19,20]. While temperatures exceeding 50°C

**Table 14. Comparison of drug temperature maintenance and anticancer hyperthermia temperature during high-temperature steam injection.**

| Range | Temperature Range (°C) | Therapeutic Effect | Literature Reference | Present Study Findings |
|---|---|---|---|---|
| Hyperthermia Range | 41.0–43.0 | Tumor suppression, induction of cell death | [34,35] | Maintained during 3–7 min interval |
| High-Temperature Zone | 44.0–48.8 | Potential tumor necrosis by heat | Based on this study | Maintained at 48.8°C during 0–2 min |
| Overheating Zone | >49.0 | Risk of normal tissue damage | [36] | Not observed in this experiment |

can damage normal tissue, no overheating (above 49°C) was observed in this experiment [36]. Fig 19 presents simulation results based on the experimentally measured temperature data, illustrating temperature changes over time following steam injection. The figure highlights the effective therapeutic range (41–43°C), the maximum recorded temperature (48.8°C), and the overheating threshold (≥49°C), demonstrating that steam injection alone can sustain clinically relevant anticancer temperatures for an extended period.

To compare the temperature changes over time following steam injection with the temperature range suitable for anticancer hyperthermia, the simulation results presented in Fig 20 are summarized in Table 15. These results demonstrate that steam injection alone can maintain temperatures within the therapeutic range for a clinically relevant duration. Importantly, no overheating-induced damage to normal tissue was observed, highlighting the potential safety and applicability of this approach for tumor treatment.

It should be noted that antibody-based drugs were not included in this experiment, as antibodies are prone to structural changes and loss of activity at high temperatures. Future research will focus on developing heat-resistant antibody formulations and designing drug delivery systems capable of preserving biological activity under elevated temperature conditions. Although high-temperature steam injection is expected to enhance tissue diffusion and promote deep drug penetration, this study did not directly analyze drug distribution or bioavailability within the tumor microenvironment. To address this, preclinical animal experiments using metastatic bone tumor models will be conducted to evaluate the thermal effects of steam-based drug delivery on tumor tissue, the extent of drug diffusion, and resulting tumor cell death.

Additionally, combination therapies incorporating antibodies or other immunotherapeutics will be investigated. Real-time monitoring of in vivo temperature changes, tissue responses, and detailed histopathological analyses will be performed to clarify the clinical applicability of this approach. Future studies will employ minipig or small animal models implanted with actual tumors to comprehensively assess drug diffusion, induction of tumor cell necrosis, heat resistance and delivery efficacy of antibody drugs, and temperature distribution within the tumor microenvironment following high-temperature steam injection. Biological response markers, including heat shock protein (HSP) expression, quantification of tumor necrosis, and immune cell infiltration, will also be analyzed to further substantiate the potential for clinical translation.

The developed high-temperature, high-pressure steam-based anticancer drug delivery system offers several advantages over existing treatments. By incorporating a temperature control module, the system maintains the target treatment temperature (120°C) with minimal fluctuations, significantly enhancing the consistency and safety of therapeutic effects. Previous studies have highlighted uncertainties in treatment outcomes due to steam temperature variability [37,38]; in contrast, our system improves the Temperature Stability Index (TSI) to 1.39% through hysteresis-based on/off control combined with an LED warning system. Additionally, drug delivery via high-temperature steam promotes greater tissue diffusion compared to conventional liquid injection methods, enabling more effective penetration into tumors [39,40]. In this study, the drug temperature was maintained at approximately 48.8°C during steam injection, corresponding to the optimal range for inducing tumor cell necrosis [5]. The system also features a simple 11-gauge needle with built-in temperature control and warning functions, enhancing ease of use and safety for clinical applications. This design is expected to improve clinical practicality relative to the complexity of existing high-temperature treatment devices [41]. Table 16

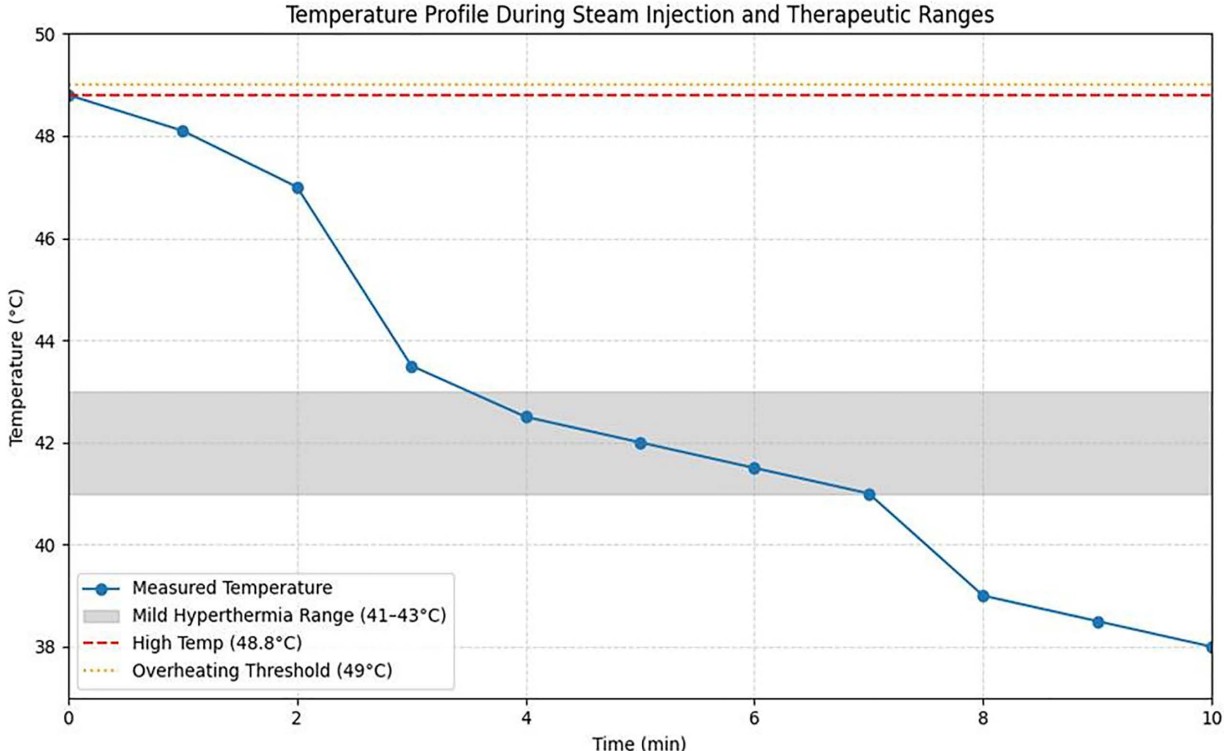

**Fig 20. Illustrates the temperature profile during high-temperature steam injection and its correlation with the known thermal therapeutic ranges.**

**Table 15. Comparison of drug temperature maintenance and therapeutic hyperthermia temperature ranges during high-temperature steam injection.**

| Range | Temperature Range (°C) | Therapeutic Effect | Literature Reference | Present Study Findings |
|---|---|---|---|---|
| Hyperthermia Range | 41.0–43.0 | Tumor suppression, induction of cell death | [34,35] | Maintained during 3–7 min interval |
| High-Temperature Zone | 44.0–48.8 | Potential tumor necrosis by heat | Based on this study | Maintained at 48.8°C during 0–2 min |
| Overheating Zone | >49.0 | Potential risk of normal tissue damage (commonly reported) | Additional references required | Not observed in this experiment |

summarizes and compares key performance indicators and characteristics of the current system with those reported in previous studies.

Finally, further discussion is warranted. This study primarily focused on evaluating the mechanical performance of a high-temperature, high-pressure steam injection system and visualizing injection patterns. Since actual anticancer drugs were not used, future studies should examine drug stability and potential degradation at 120°C. Understanding temperature-induced chemical changes and the stability of drugs in a high-temperature aqueous environment is critical for clinical application. Therefore, forthcoming research involving real anticancer agents will incorporate these essential considerations to ensure both efficacy and safety. The waveform and statistical results obtained from the software tools and measurement devices were exported as data logs and further analyzed using Python, which improved the resolution and

**Table 16. Comparison of key performance indicators and features of the developed high-temperature steam-based drug injection system with previous studies.**

| Reference | Treatment Temperature (°C) | Temperature Stability Index (TSI, %) | Drug Delivery Method | Clinical Applicability and Features |
|---|---|---|---|---|
| [This Study] | 120±1.67 | 1.39 | High-temperature Steam | 11G needle application, temperature control and warning system included |
| [37] | 115±5.3 | 5.6 | Liquid Injection | Steam hyperthermia device, large temperature variation |
| [38] | 120±4.8 | 4.8 | High-temperature Liquid Injection | Manual temperature control, lack of warning system |
| [39] | 45±2.5 | – | Low-temperature Drug Injection | Limited tumor tissue penetration |
| [40] | 50±3.1 | – | Microparticle Steam Injection | Early-stage clinical application |
| [41] | 120±6.0 | 6.2 | High-temperature Therapy Device | Complex equipment, difficult clinical application |

clarity of the results. For more detailed data and analysis procedures, please refer to the files uploaded in the Supporting Information.

## Conclusion

In this study, we designed and evaluated a system capable of minimally invasive drug delivery into bone tissue using high-temperature, high-pressure steam heated to 120°C. The system maintained a stable target temperature of 120°C through real-time monitoring with temperature sensors and a hysteresis-based temperature control algorithm, thereby ensuring consistent therapeutic efficacy and safety. To assess its performance, we conducted experiments in healthy bone tissue using a simulated mixture of fluorescent contrast agent and saline solution instead of actual anticancer drugs. The results demonstrated that high-temperature steam rapidly diffused within the tissue, maintaining both the spatial distribution of drug delivery and the therapeutic temperature over a consistent period. Compared with conventional liquid injection methods, this system significantly increased both the depth and range of drug diffusion within tissue, highlighting its potential for application in various bone-related diseases. However, the present study was limited to a small-scale animal experiment and the use of simulated substances. The current results provide preliminary evidence regarding the system's physical performance and the potential for drug diffusion. Thus, further preclinical trials employing actual anticancer drugs and tumor models are warranted. This study represents a proof-of-concept stage, and the actual clinical applicability should be verified through future tumor model experiments and clinical trials. Future work will focus on enhancing the clinical applicability of the system by improving its mechanical performance, verifying drug stability, and evaluating long-term safety. In addition, optimization studies tailored to different therapeutic settings will be undertaken.

## Supporting information

**S1 File. IACUC Approval and Compliance Documentation.**
(DOCX)

## Acknowledgments

Seon Min Lee, Kicheol Yoon, and Sangyun Lee contributed equally to this work. Seon Min Lee, Kicheol Yoon, and Sangyun Lee are the co-first (lead) authors. Hyun Guy Kang and Kwang Gi Kim are co-corresponding authors. This study was conducted with valuable advice from Gachon University Gil Medical Center (FRD2024-23-02) and the National Cancer Center (1510250−2) regarding the research environment, experimental procedures, and appropriateness of the analytical results.

## Author contributions

**Conceptualization:** Kwang Gi Kim, Kicheol Yoon, Sangyun Lee.

**Funding acquisition:** Kwang Gi Kim.

**Investigation:** Kwang Gi Kim, Kicheol Yoon, Hyun Guy Kang.

**Methodology:** Kicheol Yoon.

**Project administration:** Kwang Gi Kim.

**Software:** Seon Min Lee.

**Writing – original draft:** Seon Min Lee, Kicheol Yoon, Sangyun Lee, Hyun Guy Kang.

**Writing – review & editing:** Kicheol Yoon, Sangyun Lee, Hyun Guy Kang.

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
