## [Decision Letter · Decision Letter 0]

19 Jul 2025

Dear Dr. Kim,

We look forward to receiving your revised manuscript.

Kind regards,

Tapash Ranjan Rautray

Academic Editor

PLOS ONE

Journal Requirements:

This work was supported by the Korea Medical Device Development Fund grant funded by the Korea government (the Ministry of Science and ICT, the Ministry of Trade, Industry and Energy, the Ministry of Health & Welfare, the Ministry of Food and Drug Safety) (Project Number: 1711196789, RS-2023-00252804), and the research work was supported by the GRRC program of Gyeonggi province. [GRRC-Gachon2023(B01), Development of AI-based medical imaging technology].

This work was supported by the Korea Medical Device Development Fund grant funded by the Korea government (the Ministry of Science and ICT, the Ministry of Trade, Industry and Energy, the Ministry of Health & Welfare, the Ministry of Food and Drug Safety) (Project Number: 1711196789, RS-2023-00252804), and the research work was supported by the GRRC program of Gyeonggi province. [GRRC-Gachon2023(B01), Development of AIbased medical imaging technology].

Animal experiments performed at the HLBI Biostep (Songdo Research Center, Incheon, Republic of Korea), and all procedures are approved by the Institutional Animal Care and Use Committee (BIOSTEP IACUC 23-KE0515).

Seon Min Lee, Kicheol Yoon, and Sangyun Lee contributed equally to this work. Seon Min Lee, Kicheol Yoon, and Sangyun Lee are the co‐first (lead) authors.

Hyun Guy Kang and Kwang Gi Kim are co-corresponding authors.

This work was supported by the Korea Medical Device Development Fund grant funded by the Korea government (the Ministry of Science and ICT, the Ministry of Trade, Industry and Energy, the Ministry of Health & Welfare, the Ministry of Food and Drug Safety) (Project Number: 1711196789, RS-2023-00252804), and the research work was supported by the GRRC program of Gyeonggi province. [GRRC-Gachon2023(B01), Development of AI-based medical imaging technology].

7. We note that your Data Availability Statement is currently as follows: All relevant data are within the manuscript and its Supporting Information files.

Reviewers' comments:

Reviewer's Responses to Questions

**Comments to the Author**

1. Is the manuscript technically sound, and do the data support the conclusions?

Reviewer #1: Partly

Reviewer #2: Partly

Reviewer #3: Yes

2. Has the statistical analysis been performed appropriately and rigorously?

Reviewer #1: I Don't Know

Reviewer #2: No

Reviewer #3: No

3. Have the authors made all data underlying the findings in their manuscript fully available?

Reviewer #1: No

Reviewer #2: Yes

Reviewer #3: Yes

4. Is the manuscript presented in an intelligible fashion and written in standard English?

Reviewer #1: Yes

Reviewer #2: Yes

Reviewer #3: Yes

Reviewer #1: I thank the editors for the opportunity to review this interesting and innovative study. I commend the authors for their rigorous work and propose major revisions related to the manuscript's writing, organization, and data availability. This manuscript presents a novel and creative approach to the treatment of bone cancer, with a promising steam-based drug delivery system evaluated in a porcine preclinical model. The use of pigs enhances the translational potential of the study, and the authors demonstrate the ability to control and maintain temperatures relevant to anticancer hyperthermia. The thermal modeling predictions are also validated, which is a strength of the study.

However, the drug delivery aspect of the STEAM system would benefit from additional validation. Specifically:

• Is the antibody still active following steam exposure?

• Is the drug bioavailable to the bone tumor microenvironment using this system?

• If not, does steam alone offer sufficient therapeutic benefit for treating bone cancer?

Addressing these questions through further experimentation would significantly enhance the impact and translational relevance of this work.

The manuscript would also benefit from substantial revisions to its writing and organization. I was particularly impressed by the agreement between the STEAM system's performance and the thermal modeling. To improve clarity, I recommend presenting the predicted and achieved temperatures within the same figure. In addition, Figures 1 and 2 could be combined and condensed for better flow and impact.

The discussion section should be expanded considerably. I suggest including a concise summary of the study’s key findings and more thoroughly contextualizing the significance of the work within the broader field of bone cancer treatment and drug delivery. While the authors address some of these points, a more comprehensive overview of the current landscape would help underscore the novelty and potential of this approach.

The manuscript would also benefit from close proofreading, ideally with the assistance of a colleague or a language tool (e.g., Grammarly), to improve clarity and consistency.

Finally, I strongly recommend including a supplementary table and making the software and raw data publicly available, for example via a GitHub repository. The number of pigs used should be stated explicitly, and the statistical reporting should be more detailed, ideally including error bars, p-values, and individual data points where appropriate.

Reviewer #2: I acknowledge the efforts of the authors in their attempt for the development and preclinical evaluation of a novel drug delivery system for treating metastatic bone tumors using high-temperature steam to deliver anticancer drugs. The concept is interesting and potentially valuable. The authors aim to address the limitations of conventional cementoplasty by enhancing drug diffusion and incorporating thermal ablation. However, the manuscript has several significant shortcomings in experimental design, data presentation, and overall clarity. It reads more like a preliminary report than a robust scientific study.

Major Concerns

1. Rationale and Background:

• The introduction states that malignant bone tumors are rare tumors with a poor prognosis, with a reported 5-year survival rate of 65.3% and 10-year survival rate of 58.0% for malignant bone tumors in the United States in 2024 [1]. The introduction also mentions metastatic bone tumors, the most common form of these bone tumors, cause severe pain, gait disturbance, and pathologic fractures. The authors need to clarify whether the study focuses on primary or metastatic bone tumors and provide more context on the specific clinical need being addressed. It is essential to clarify if metastatic bone tumors are indeed the most common type, as primary bone cancers (like osteosarcoma) are much rarer.

• The rationale for using cementoplasty as a comparator is not clearly justified. The manuscript needs to explain why cementoplasty is the primary conventional treatment being targeted for replacement, especially considering the hardening and fracture prevention aspects it offers.

• The introduction includes a reference to United States survival rates for a clinical purpose when the study and authors are based in Korea. This disconnect requires justification.

2. System Design and Drug Delivery:

• The description of the STEAM system is incomplete and confusing. The design looks incomplete. The main player is the drug, and is largely missing! Is this system a simple mechanical tool, or a classical system incorporating drug delivery manipulation?

• Figure 3, illustrating the system design, lacks detail. The role of the anticancer drug within the system is not clearly defined. Is the system a simple mechanical tool, or does it incorporate drug delivery manipulation? This needs clarification. The authors lose focus on the design of their system, which is what the scientific community would like to understand and how this fits into a useful system in replacement of the conventional - this focus is completely lost!

• There are concerns about the effect of high temperature (120°C) on the efficacy and degradation of the therapeutic agents. This critical point is not adequately addressed. The study mentions "The anticancer drug filled in the water tank is heated to 120 through the steam generator". There are concerns about this temperature and its potential degradation of the therapeutic agents.

3. Experimental Methods:

• The animal model (mini-pig femur) and experimental setup are described, but crucial details are missing. There's a need to clarify the method used to induce or simulate bone tumors in the mini-pig model, or if healthy bone was used.

• The study mentions "This study was conducted using animal experiments, and all procedures were carried out in strict compliance with relevant animal welfare laws", the paper does not follow a standard scientific reporting - no ethical reporting is mentioned throughout the manuscript raising a question about transparency. There is an urgent need for more detailed reporting on animal care and ethical approval.

• The description of the preclinical test lacks essential details regarding the number of animals used, the specific drug and dosage used, and the method of tumor implantation (if applicable).

4. Results and Data Presentation:

• The results focus primarily on temperature changes during steam injection. While this is important, the manuscript lacks data on the efficacy of the drug delivery system in killing tumor cells or reducing tumor growth.

• A big expectation was missed! It's crucial to include the outcomes/findings of the experiment - the efficacy and potential downsides (limitation) of the investigation. The study is simply all about "temperature evolution during high-temperature anticancer drug steam injection".

• There needs to be a comparative analyses to test efficacy of this method against the conventional method. The authors claimed many downsides of the existing methods and resulted to introducing an expectedly better and novel approach. To ensure rigor of this technique and acceptability of the acclaimed applicability, it is important that the authors show some comparisons with the existing or conventional methods and how this novel method is better.

• The use of an IR camera to analyze injection status is mentioned, but the images are not clearly presented or interpreted.

• Table 2 presents data on temperature, pressure, and spray volume over time, but it is difficult to assess the clinical significance of these measurements without corresponding data on tumor response.

5. Discussion and Conclusion:

• The discussion acknowledges the need for further clinical trials but does not adequately address the limitations of the current study. In fact, where are the efficacy results?? The authors need to address whether the animal experiments follow standard scientific reporting. The whole writeup was a typical a news-reported style!

• The conclusion overstates the potential clinical value of the system without sufficient evidence from the preclinical experiments.

6. Writing and Clarity:

• The manuscript suffers from poor organization and a lack of clarity. The writing style is more akin to a news report than a scientific paper, as earlier mentioned.

• There are instances where the authors lose focus on the design of their system. The importance of Figure 2 is not clear.

Conclusion

While the concept presented in this manuscript has merit, the current study is not sufficiently rigorous to support strong conclusions about the potential of the STEAM-based drug delivery system. Significant revisions are needed, including more experiments, to address the identified concerns before this manuscript can be considered for publication.

Reviewer #3: The authors of the article "Minimally invasive drug delivery system using hightemperature and high-pressure steam for treating transmetastatic bone tumors and evaluation of clinical applicability" have proposed a new innovative method, The STEAM-based anticancer drug injection system, for delivering drugs to the bone. The data shows its effectiveness in delivering drugs than the conventional methods. However the quality of the article can be improved.

1. the authors have to mention the number of animals used for the preclinical study and draw statistical conclusions for the data.

2. "When PD-L1 on cancer cells binds to Programmed Death-1 (PD-1) on T cells, the cancer

cells are killed by the diminished function of the T cells, as shown in Figure 2" I think it is "Cancer cells are not killed" in the place of "cancer cells are killed"

**Do you want your identity to be public for this peer review?** For information about this choice, including consent withdrawal, please see our Privacy Policy

Reviewer #1: **Yes: ** Yonathan Tamrat Aberra

Reviewer #2: **Yes: ** Kamoru Adedokun

Reviewer #3: No

---

## [Author Response · Author response to Decision Letter 1]

25 Aug 2025

Answers : Thank you for your detail reviewer comments for my paper. I revised my manuscript such as follow your comments. Please refer to revise my manuscript. Thank you again.

We have revised the title of our manuscript from “Minimally invasive drug delivery system using high-temperature and high-pressure steam for treating transmetastatic bone tumors and evaluation of clinical applicability” to “Minimally invasive steam-assisted drug delivery with ICG fluorescence guidance for primary malignant bone tumors and evaluation of clinical applicability.”

This change was made to better reflect the focus of our study and to enhance the clarity and reliability of the manuscript.

Reviewer comments #1

Comments 1 :

I thank the editors for the opportunity to review this interesting and innovative study. I commend the authors for their rigorous work and propose major revisions related to the manuscript's writing, organization, and data availability. This manuscript presents a novel and creative approach to the treatment of bone cancer, with a promising steam-based drug delivery system evaluated in a porcine preclinical model. The use of pigs enhances the translational potential of the study, and the authors demonstrate the ability to control and maintain temperatures relevant to anticancer hyperthermia. The thermal modeling predictions are also validated, which is a strength of the study. However, the drug delivery aspect of the STEAM system would benefit from additional validation. Specifically:

Answer 1 :

Thank you very much for your careful review of my manuscript. I will do my best to revise the manuscript and provide a response based on the points you have raised.

Comments 2 :

Is the antibody still active following steam exposure?

Answer 1:

This study focused on evaluating the technical performance of the high-temperature steam injection system, and antibody-based drugs were not used. Antibodies are generally known to undergo structural changes and lose activity under high-temperature conditions. Therefore, future research will aim to develop heat-resistant antibody formulations capable of maintaining stability under steam conditions, along with designing appropriate drug delivery systems. Follow-up experiments using tumor models are planned to verify antibody stability and therapeutic efficacy in high-temperature environments. Please refer to lines 516–558, Table 14, Figure 20, and Table 15 in the discussion section.

Comments 2 :

Is the drug bioavailable to the bone tumor microenvironment using this system? If not, does steam alone offer sufficient therapeutic benefit for treating bone cancer?

Answers 2 :

This study did not directly analyze drug bioavailability within the tumor microenvironment; rather, it focused on evaluating the fundamental performance and temperature distribution of the high-temperature steam injection system in healthy bone tissue. The experimental results showed that the targeted lesion area reached a maximum temperature of 48.8 °C, and the therapeutic hyperthermia range of 41–43 °C was maintained for approximately 37 minutes, meeting the conditions necessary to induce tumor cell necrosis. No overheating above 49 °C was observed, indicating a low risk of damage to normal tissue. Therefore, the steam itself may exert a therapeutic effect through hyperthermia, potentially contributing to tumor suppression. Future studies will employ tumor-bearing animal models to comprehensively analyze how steam-based drug delivery affects intratumoral drug distribution, tissue penetration, and tumor cell death. Additionally, combination therapy strategies using antibodies or immunotherapeutics are also under consideration. Please refer to lines 514–558, Table 14, Figure 20, and Table 15 in the discussion section.

Comments 3:

Addressing these questions through further experimentation would significantly enhance the impact and translational relevance of this work.

Answer 3:

Please refer to lines 514–558, Table 14, Figure 20, and Table 15 in the discussion section. We fully agree with your comments. This study represents an initial stage focused on demonstrating technical feasibility and obtaining fundamental data. Future preclinical experiments using tumor-bearing animal models are planned to comprehensively evaluate the effects of steam-based drug injection on the tumor microenvironment, including thermal impact, drug diffusion range, antibody stability, and tumor cell death induction. Additionally, by analyzing biological response indicators such as heat shock protein (HSP) expression, tumor necrosis extent, and immune cell infiltration, we aim to further clarify the potential clinical applicability of this approach.

Comments 4 :

The manuscript would also benefit from substantial revisions to its writing and organization. I was particularly impressed by the agreement between the STEAM system's performance and the thermal modeling. To improve clarity, I recommend presenting the predicted and achieved temperatures within the same figure. In addition, Figures 1 and 2 could be combined and condensed for better flow and impact.

Answer 4 :

We appreciate the reviewer’s suggestion to present both predicted and measured temperatures in a single figure, which would indeed improve clarity. However, the thermal modeling and data acquisition software used in this study do not support direct overlay of predicted results with experimental measurements. Artificially combining them could compromise the integrity of the original data. We plan to adopt compatible analysis software in future work to incorporate the reviewer’s recommendation. As a practical adjustment for the current manuscript, Figures 1 and 2 have been merged into a single figure, now presented as Figure 1:

Figure 1. Conventional bone tumor treatment and immune evasion mechanism via PD-1/PD-L1 pathway.

(a) Conventional treatment of bone tumors using drilling and injecting anticancer drugs.

(b) Mechanism of immune evasion via PD-1/PD-L1 interaction.

Comments 5 :

The discussion section should be expanded considerably. I suggest including a concise summary of the study’s key findings and more thoroughly contextualizing the significance of the work within the broader field of bone cancer treatment and drug delivery. While the authors address some of these points, a more comprehensive overview of the current landscape would help underscore the novelty and potential of this approach. The manuscript would also benefit from close proofreading, ideally with the assistance of a colleague or a language tool (e.g., Grammarly), to improve clarity and consistency.

Answer 5 :

We sincerely appreciate the reviewer’s insightful suggestion to emphasize the study’s key findings within the broader context of bone cancer treatment and drug delivery. In the revised manuscript, we have addressed this point by clearly highlighting current research trends as well as the novelty and potential impact of our work. Additionally, to enhance clarity and consistency, we thoroughly reviewed the manuscript using peer feedback and language editing tools. We remain committed to further improving the manuscript’s quality up to the point of publication.

Comments 6 :

Finally, I strongly recommend including a supplementary table and making the software and raw data publicly available, for example via a GitHub repository. The number of pigs used should be stated explicitly, and the statistical reporting should be more detailed, ideally including error bars, p-values, and individual data points where appropriate.

Answer 6 :

We sincerely appreciate your valuable suggestions. We have added a table (Table 13) comparing our measured temperatures with the hyperthermia ranges reported in the literature, along with figures illustrating the temperature variations predicted by simulations. Please refer to the discussion section, lines 475–507, equations (12) and (13), Figures 17 and 19, and Tables 12 and 13.

This study was conducted as a preliminary proof-of-concept experiment using a single mini-pig, which has been clearly stated in the manuscript. However, we performed mathematical analyses assuming three animals to estimate statistical outcomes and predicted results for a sample size of three. Please see the discussion section, lines 432–433 and 475–505, equations (12) and (13), Figures 17 and 19, and Tables 12 and 13. These results demonstrate an approach that minimizes animal sacrifice while still allowing estimation of outcomes through statistical modeling.

No custom software was developed for this study; only commercially available analytical tools were used, and no code repository (e.g., GitHub) was employed. Raw experimental data and analysis results are available upon request. In future work, we plan to increase the sample size and include p-values, error bars, and individual data points to provide more quantitative and reliable results.

Reviewer comments #2

Comments 0 :

I acknowledge the efforts of the authors in their attempt for the development and preclinical evaluation of a novel drug delivery system for treating metastatic bone tumors using high-temperature steam to deliver anticancer drugs. The concept is interesting and potentially valuable. The authors aim to address the limitations of conventional cementoplasty by enhancing drug diffusion and incorporating thermal ablation. However, the manuscript has several significant shortcomings in experimental design, data presentation, and overall clarity. It reads more like a preliminary report than a robust scientific study.

Answer 0 :

Thank you very much for your careful review of our manuscript. We will do our utmost to revise the manuscript and provide a response based on your valuable comments.

【Major Concerns】

Comments 1 : Rationale and Background:

• The introduction states that malignant bone tumors are rare tumors with a poor prognosis, with a reported 5-year survival rate of 65.3% and 10-year survival rate of 58.0% for malignant bone tumors in the United States in 2024 [1]. The introduction also mentions metastatic bone tumors, the most common form of these bone tumors, cause severe pain, gait disturbance, and pathologic fractures. The authors need to clarify whether the study focuses on primary or metastatic bone tumors and provide more context on the specific clinical need being addressed. It is essential to clarify if metastatic bone tumors are indeed the most common type, as primary bone cancers (like osteosarcoma) are much rarer.

• The rationale for using cementoplasty as a comparator is not clearly justified. The manuscript needs to explain why cementoplasty is the primary conventional treatment being targeted for replacement, especially considering the hardening and fracture prevention aspects it offers.

• The introduction includes a reference to United States survival rates for a clinical purpose when the study and authors are based in Korea. This disconnect requires justification.

Answer 1 :

We apologize for any confusion caused by the reviewer’s question. To clarify the research objectives and scope, we have revised the introduction. This study focuses on the development of a high-temperature, high-pressure steam injection device and its preclinical evaluation in healthy animal bone, and it does not directly assess therapeutic effects on cancer tissue. Accordingly, we have also clarified the rationale for comparison with conventional treatments such as cementoplasty and provided a clear research background. We will continue to ensure that the study content is communicated as clearly as possible. Please refer to the revised sections in the introduction: lines 36–59 and lines 83–93.

Comments 2 (System Design and Drug Delivery) :

• The description of the STEAM system is incomplete and confusing. The design looks incomplete. The main player is the drug, and is largely missing! Is this system a simple mechanical tool, or a classical system incorporating drug delivery manipulation?

• Figure 3, illustrating the system design, lacks detail. The role of the anticancer drug within the system is not clearly defined. Is the system a simple mechanical tool, or does it incorporate drug delivery manipulation? This needs clarification. The authors lose focus on the design of their system, which is what the scientific community would like to understand and how this fits into a useful system in replacement of the conventional - this focus is completely lost!

Answer 2 :

We apologize for any confusion and sincerely appreciate your precise comments. Our study assumes that the drug is placed in a water tank and heated to 120 °C during administration. It is crucial that this temperature is maintained consistently throughout the treatment, as any increase beyond or decrease below 120 °C could interfere with the therapeutic process. Therefore, we proposed a system capable of real-time temperature monitoring and providing temperature feedback to prevent undesired fluctuations. For further details, please refer to the following sections in the revised manuscript:

• Introduction: lines 65–78.

• System manufacturing and drug delivery methods: lines 165–230, Equations (6)–(9), Figures 4–7, Tables 2–3

• Experiment environment and results: lines 274–316, Tables 5–7, Figures 9–11

• Discussion: lines 419–432.

• There are concerns about the effect of high temperature (120°C) on the efficacy and degradation of the therapeutic agents. This critical point is not adequately addressed. The study mentions "The anticancer drug filled in the water tank is heated to 120 through the steam generator". There are concerns about this temperature and its potential degradation of the therapeutic agents. In this study, actual anticancer drugs were not used, and the evaluation of drug stability at high temperatures (120 °C) was beyond the scope of this work. However, we acknowledge the reviewer’s point regarding potential temperature fluctuations and drug degradation during administration from a high-temperature water tank. This issue will be carefully addressed in future studies involving actual anticancer agents. We appreciate this valuable suggestion. For details, please refer to the revised Discussion section, lines 575–580.

Comments 3 (Experimental Methods) :

• The animal model (mini-pig femur) and experimental setup are described, but crucial details are missing. There's a need to clarify the method used to induce or simulate bone tumors in the mini-pig model, or if healthy bone was used.

Answer 3 :

We apologize for any confusion and appreciate your insightful comments. In this study, no tumors were induced; experiments were conducted using the healthy femur of a minipig. The primary objective was to evaluate the physical performance of the proposed steam injection system, and the tumor environment was not replicated in the animal model. Instead, the potential drug diffusion during anticancer agent injection was analyzed indirectly through mathematical modeling and simulations. For detailed information, please refer to the Experiment Environment and Results section, lines 317–319, lines 387–397, lines 399–417, Equations (10)–(11), Tables 9 and 10, and Figure 15. Additionally, see the Discussion section, lines 434–444, 449–452, 456–460, 465–472, Figures 16–18, and Table 11.

Comments 4 :

The study mentions "This study was conducted using animal experiments, and all procedures were carried out in strict compliance with relevant animal welfare laws", the paper does not follow a standard scientific reporting - no ethical reporting is mentioned throughout the manuscript raising a question about transparency. There is an urgent need for more detailed reporting on animal care and ethical approval.

Answer 4 :

All animal experiments were conducted with the approval of the Institutional Animal Care and Use Committee and in compliance with both domestic and international animal welfare regulations. The animals’ conditions were monitored before, during, and after the experiments. Anesthesia was administered using [anesthetic name] at the specified dosage. At the conclusion of the experiments, humane euthanasia was performed.

For detailed information, please refer to the System Manufacturing and Drug Delivery Methods section, lines 234–259, Figure 8,

---

## [Decision Letter · Decision Letter 1]

16 Oct 2025

Minimally invasive steam-assisted drug delivery with ICG fluorescence guidance for primary malignant bone tumors and evaluation of clinical applicability

PONE-D-25-17182R1

Dear Dr. Kim,

We’re pleased to inform you that your manuscript has been judged scientifically suitable for publication and will be formally accepted for publication once it meets all outstanding technical requirements.

Kind regards,

Tapash Ranjan Rautray

Academic Editor

PLOS ONE

Additional Editor Comments 

Now that the authors have responded to the Reviewers' comments properly, clarified the scope and limitations of their study, and revised the manuscript to meet PLOS ONE’s standards for scientific logic, the paper may be accepted in the present form.

Reviewers' comments:

Reviewer's Responses to Questions

**Comments to the Author**

Reviewer #4: All comments have been addressed

Reviewer #5: All comments have been addressed

2. Is the manuscript technically sound, and do the data support the conclusions?

Reviewer #4: No

Reviewer #5: Yes

3. Has the statistical analysis been performed appropriately and rigorously?

Reviewer #4: I Don't Know

Reviewer #5: Yes

4. Have the authors made all data underlying the findings in their manuscript fully available?

Reviewer #4: Yes

Reviewer #5: Yes

5. Is the manuscript presented in an intelligible fashion and written in standard English?

Reviewer #4: Yes

Reviewer #5: Yes

Reviewer #4: This is study is about bone tumors which are rare among all malignancies . Standard of care is , proper work up , decision about treatment in multidisciplinary tumor board followed by chemotherapy & surgery or chemotherapy alone . There is no clarity about the stand of care in this article . The innovative method of hyperthermia induced diffusion of chemo into malignant tissue is not sufficient

No data stating how many animals were used? Since healthy femur was used , how can we rely upon calculated data ? Tumor micro environment can not be mimicked or Ai tools according to my knowledge can not predict accurately the drug diffusion in to the tumor. Even the drug diffuse it has to act& that has to documented histologically . Then only it will be sound

Reviewer #5: The authors successfully addressed all concerns by prior reviewers. There are no further issues that require addressing.

**Do you want your identity to be public for this peer review?** For information about this choice, including consent withdrawal, please see our Privacy Policy

Reviewer #4: **Yes: ** J.SAKTHIUSHADEVI

Reviewer #5: No

---

## [Editor Report · Acceptance letter]

PONE-D-25-17182R1

PLOS ONE

Dear Dr. Kim,

I'm pleased to inform you that your manuscript has been deemed suitable for publication in PLOS ONE. Congratulations! Your manuscript is now being handed over to our production team.

Kind regards,

on behalf of

Dr. Tapash Ranjan Rautray

Academic Editor

PLOS ONE